# Long-Context LLMs Meet RAG: Overcoming Challenges for Long Inputs in RAG

**Bowen Jin**[1][§]**, Jinsung Yoon**[2]**, Jiawei Han**[1]**, Sercan Ö. Arık**[2]
[1]University of Illinois Urbana-Champaign  [2]Google Cloud
{bowenj4,hanj}@illinois.edu, {jinsungyoon,soarik}@google.com

## Abstract

Retrieval-augmented generation (RAG) empowers large language models (LLMs) to utilize external knowledge sources. The increasing capacity of LLMs to process longer input sequences opens up avenues for providing more retrieved information, to potentially enhance the quality of generated outputs. From a long-context LLM perspective, it assumes that a larger retrieval set would contain more relevant information (higher recall), that might result in improved performance. However, our empirical findings demonstrate that for many long-context LLMs, the quality of generated output initially improves first, but then subsequently declines as the number of retrieved passages increases. This paper investigates this phenomenon, identifying the detrimental impact of retrieved "hard negatives" as a key contributor. To mitigate this and enhance the robustness of long-context LLM-based RAG, we propose both training-free and training-based approaches. We first showcase the effectiveness of retrieval reordering as a simple yet powerful training-free optimization. Furthermore, we explore training-based methods, specifically RAG-specific implicit LLM fine-tuning and RAG-oriented fine-tuning with intermediate reasoning, demonstrating their capacity for substantial performance gains. Finally, we conduct a systematic analysis of design choices for these training-based methods, including data distribution, retriever selection, and training context length.

## 1 Introduction

Retrieval-augmented generation (RAG) (Gao et al., 2023) empowers large language models (LLMs) to utilize external information sources by selecting the relevant pieces from a large corpus (Zhao et al., 2023), thereby enhancing their effectiveness, customizability and efficiency in complex problem-solving (Yu et al., 2024a; Chen et al., 2024b). RAG can also mitigate issues such as factual inaccuracies (Augenstein et al., 2023) and hallucinations (Huang et al., 2023), which LLMs often exhibit when confronted with knowledge-intensive tasks. RAG systems typically employ a retriever to identify relevant information from a corpus, which is then presented in the context of an LLM as the generator.

Recent advances in computational resources and methodological innovations have enabled the development of LLMs that support increasingly longer context (Reid et al., 2024; Dubey et al., 2024). This has even opened up new avenues for directly inputting entire corpora or knowledge bases into the LLMs. Yet, it would still not be feasible for large corpora (*e.g.*, Wikipedia) and can incur higher computational costs. Despite extensive research on RAG (Xu et al., 2023; Li et al., 2024; Lee et al., 2024), the interplay with long-context LLMs, particularly how to optimally design RAG systems using them effectively, remains under-explored. Existing works (Lin et al., 2024; Asai et al., 2024; Yoran et al., 2024) propose tuning LLMs for RAG, but predominantly focus on a limited number of retrieved passages (< 10). From the long-context LLM perspective (Reid et al., 2024; Dubey et al., 2024), longer context would allow for the inclusion of more retrieved passages, leading to higher recall and potentially improved performance. However, our findings reveal that this does not always hold true and highlight the need for a careful re-evaluation of standard RAG designs when utilizing long-context LLMs. We demonstrate that achieving optimal performance in such systems and to fully utilize the opportunities provided by the LLMs require a holistic rethinking and effective novel approaches to the unique challenges.

---

[§]Work was done while Bowen was a student researcher at Google Cloud.

Though there are some studies on evaluating long-context LLMs in synthetic benchmarks such as needle-in-the-haystack (Kamradt, 2023), their capability in real long-context RAG scenarios is underexplored. This paper presents comprehensive analyses on long-context LLMs in RAG systems. Contrary to the suggestions of previous work (Xu et al., 2023; Li et al., 2024), our research reveals that increasing the number of retrieved passages does not consistently improve performance with long-context LLMs (Section 3.1). Instead, we observe that the generative modeling performance initially increases and then declines – simply providing more retrieved passages does not guarantee better outcomes. Using stronger retrievers is also not a mitigation mechanism – indeed the performance degradation can even be more severe with them. For deeper understanding of the phenomenon, we conduct further investigations, which reveal that increasing the number of retrieved passages can introduce irrelevant information ("noise") that can mislead the LLM generation (Section 3.2). We also examine the impact of "hard negatives" of different retrievers on the LLMs, and show that there are scenarios where the 'hard negatives' from stronger retrievers might confuse the LLM generation even more than those from weaker retrievers (Section 3.3).

To address the challenges identified in our analyses, we propose three methods, encompassing both training-free and training-based approaches, to enhance the performance of long-context LLMs in RAG applications: (1) Retrieval reordering: recognizing the "lost-in-the-middle" phenomenon observed for long-context LLMs (Liu et al., 2024), we propose reordering retrieved documents based on their retrieval scores. By prioritizing documents with higher scores at the beginning and end of the input sequences, we guide the LLMs' attention towards more relevant information and mitigate the impact of hard negatives. (2) Implicit robustness fine-tuning: given the ability to handle noisy retrieved context is not explicitly acquired during standard LLM training, we propose tuning the LLMs with the data comprising queries and retrieved documents, including those with potential noise. This encourages the LLMs to implicitly learn robustness to hard negatives. (3) Explicit relevance fine-tuning: while the previous method implicitly enhances robustness, it does not explicitly teach the LLMs to identify relevant documents. Therefore, we propose augmenting the LLM tuning with an intermediate reasoning step, where the LLMs are trained to analyze the retrieved documents and explicitly identify relevant information before generating the final output. This approach aims to improve the LLMs' ability to discern relevant information from noise within the retrieved context.

Overall, the main contributions can be summarized as follows:

- Systematic analysis of long-context RAG: we systematically analyze the use of long-context LLMs in RAG systems, specifically (1) examining the conflict impact of retrieved "relevant documents" and "hard negatives" on RAG performance when scaling up the number of retrieved passages and (2) exploring the correlation between retriever strength and hard negative difficulty.

- Novel methods for robust RAG: we propose three methods to improve the robustness of long-context LLMs in RAG: (1) a training-free method based on retrieval reordering, (2) implicit tuning for robustness to hard negatives and (3) explicit tuning with intermediate reasoning for relevance identification. Overall, our proposed approaches show significant accuracy and robustness improvements on long-context RAG performance.

- Comprehensive study of RAG-specific LLM tuning: we conduct a thorough investigation into various factors influencing the effectiveness of RAG-specific tuning, including data distribution, the employed retriever, and training context length.

## 2 RELATED WORK

Large language models (LLMs) can be prone to hallucinations especially at knowledge-intensive tasks (Zhao et al., 2023; Huang et al., 2023; Augenstein et al., 2023). Retrieval-augmented generation (RAG) addresses this by incorporating external knowledge sources to provide accurate and relevant information (Gao et al., 2023). Traditional RAG systems comprise a retriever to identify relevant information and a generator to synthesize the answer (Zhao et al., 2024; Zhu et al., 2021). While previous research focused on improving either the retriever (Karpukhin et al., 2020; Izacard et al., 2021; Wang et al., 2022) or the generator (Dong et al., 2022; Liu et al., 2024; Agarwal et al., 2024) in isolation, we take a holistic approach. Conducting comprehensive analyses of the entire RAG system, we focus on the challenges and opportunities presented by using long-context LLMs as generators. We propose novel solutions to better employ them in long-context RAG.

Increased computational resources and advancements in efficient training methods have pushed LLMs supporting longer inputs (Wang et al., 2024; Zhou et al., 2024). While long-context LLMs (Reid et al., 2024) demonstrated impressive performance on benchmarks like "needle-in-the-haystack" (Kamradt, 2023) and RULER (Hsieh et al., 2024a), these benchmarks often rely on random negative examples and do not accurately reflect the challenges posed by the "hard negatives" encountered in real-world RAG scenarios (Cuconasu et al., 2024). Furthermore, existing studies on long-context LLMs in multi-document settings (Liu et al., 2024; Shi et al., 2023) often assume a single "golden" document and random negatives, which differs from the RAG context where multiple relevant passages and hard negatives may exist (Hsieh et al., 2024b; Cuconasu et al., 2024). Although some research has explored the relationship between RAG and long-context LLMs (Xu et al., 2023; Li et al., 2024; Lee et al., 2024), these works take different perspectives. They mainly focus on studying the (1) trade-offs between RAG and long-context LLMs (Xu et al., 2023), (2) routers to manage RAG and long-context LLMs (Li et al., 2024), (3) and the potential for LLMs to replace retrieval entirely (Lee et al., 2024), while leaving long-context LLMs as generators in RAG under-explored. Some existing work (Yu et al., 2023; Yan et al., 2024) identify the harm of irrelevant retrieved information in RAG, but they mainly focus on synthetic scenario where no more relevant information is added, which is different from the real-world RAG systems with large-scale retrieval. We delve deeper into the potential benefits of long-context LLMs for RAG and investigate how to optimize these LLMs specifically for this application.

Previous research has explored adapting LLMs for RAG using instruction tuning (Zhang et al., 2023). RetRobust (Yoran et al., 2024) fine-tunes LLMs with 1 retrieved relevant passage or random negative passage to make it robust to irrelevant passage. RA-DIT (Lin et al., 2024) conducts dual instruction tuning to make the LLM more effectively leverage retrieved information and retriever provide results more aligned with LLM preference. Self-RAG (Asai et al., 2024) introduces a framework to train a LM that dynamically retrieves passages, generates content, and evaluates the retrieved passages for improved performance. RAFT (Zhang et al., 2024) trains the LLMs to improve their ability to answer questions in "open-book" in-domain settings. More recently, RankRAG (Yu et al., 2024b) tunes a LLM for the dual purpose of context ranking and answer generation in RAG. InstructRAG (Wei et al., 2024) finetunes the LLM to generate self-synthesized rationales rather than directly answering the question. However, these existing efforts primarily focus on tuning with a limited number of retrieved passages (typically fewer than 10) and do not fully leverage the potential of long-context LLMs. This work aims to address this gap by specifically investigating how to optimize long-context LLMs for large-scale RAG, where the number of retrieved passages can be significantly higher.

## 3 Challenges of Long context LLMs in RAG

We present a systematic investigation into the challenges of utilizing long-context LLMs in RAG. Each subsection focuses on a specific research question, outlining corresponding experiments and analyzing the results on the key challenges. These insights inform the development of targeted solutions for improving RAG performance with long-context LLMs, which are presented in subsequent sections.

### 3.1 The Effect of retrieved context size on RAG performance

This subsection investigates the relationship between the number of retrieved passages and the performance of long-context LLMs in RAG systems.

**Research question.** Long-context LLMs offer the potential to incorporate more retrieved passages into RAG systems. This raises a crucial question: *Does a larger volume of retrieved context consistently translate to better performance when using long-context LLMs in RAG?*

**Experimental setting.** We evaluate the performance of RAG systems on the Natural Questions (NQ) (Kwiatkowski et al., 2019) dataset using two different retrievers (BM25 (Robertson et al., 2009) and e5 (Wang et al., 2022), where e5 exhibits higher performance on NQ (Recall@40 is 0.90 with e5 and 0.73 with BM25)) and four long-context LLMs (Gemma-7B-Chat (Team et al., 2024a), Gemma-2-9B-Chat (Team et al., 2024b), Mistral-Nemo-12B-Instruct (Jiang et al., 2023) and Gemini-1.5-pro (Reid et al., 2024)). We systematically vary the number of passages retrieved by each retriever.

**Observations.** Figure 1 presents the following key observations: 1) Strong Retriever (e5): Across all LLMs, increasing the number of retrieved passages initially improves performance, but then

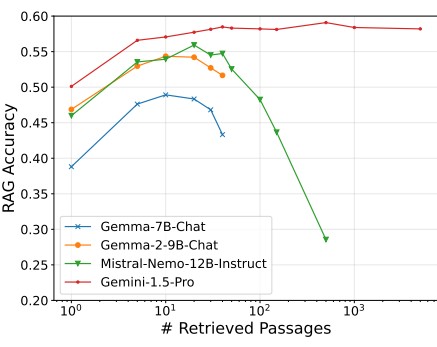 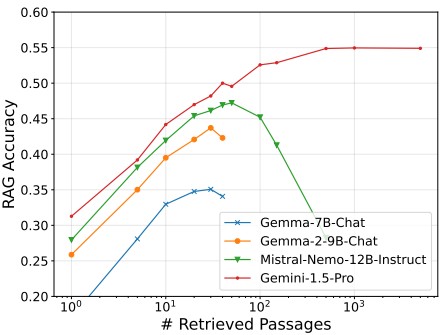

(a) RAG performance with e5 retriever     (b) RAG performance with BM25 retriever

Figure 1: Impact of retrieved context size on RAG performance with 4 different LLMs on NQ. Increasing the number of retrieved passages initially improves performance but then leads to a decline. This degradation is more pronounced using a retriever (e5) that exhibits higher recall@k on NQ compared to BM25 (Recall@40 is 0.90 with e5 and 0.73 with BM25). The maximum number of retrieved passages varies across LLMs due to differences in their maximum token limits.

leads to a sharp decline or plateau. 2) Weak Retriever (BM25): Performance generally exhibits a continuous increase or a slight decrease as the number of retrieved passages increases. While these observations may appear counter-intuitive - given that one might expect monotonic improvements due to higher recall (*i.e.*, a greater chance of retrieving relevant information) - the inclusion of additional documents can reduce precision, with irrelevant or misleading passages detracting LLMs from overall performance. Comparison of different retrievers, the results on other datasets and case studies are shown in Appendix A, B.1/R and O.

**Insights.** The effectiveness of increasing retrieved context size in RAG depends on the strength of the retriever. With both retrievers, performance exhibits an "inverted-U pattern". However, the curve with a strong retriever reaches the performance peak earlier and the decreases faster than that with a weak retriever. This suggests that factors beyond the amount of retrieved information are at play.

## 3.2 THE INTERPLAY OF RETRIEVAL QUALITY AND LLM CAPABILITIES

This subsection delves into the factors hindering the performance of long-context LLMs in RAG, aiming to discern whether limitations arise from retrieval quality or the LLM's ability to process the retrieved information.

**Research question.** *Do the observed performance bottlenecks originate from limitations in the retriever's ability to identify relevant information, or from the long-context LLM's capacity to effectively utilize the retrieved context?*

**Experimental setting.** We analyze the relationship between RAG performance and retrieval quality, specifically recall and precision, using the Gemma-2-9B-Chat LLM with both e5 and BM25 retrievers (Figure 2). Recall@k measures the presence of relevant passages within the top-k retrieved passages, while precision@k quantifies the proportion of relevant passages among them.

**Observations.** Increasing the number of retrieved passages consistently leads to higher recall but lower precision, irrespective of the retriever used. Crucially, the overall accuracy of the RAG system falls below the recall across all retrieval sizes. This indicates that even when relevant information is present in the retrieved context, the LLM may fail to generate the correct answer. This demonstrates that the irrelevant retrieved passages can sometimes mislead the LLM. Furthermore, despite exhibiting higher precision, the e5 retriever leads to a more pronounced performance degradation as the number of retrieved passages increases compared to BM25.

**Insights.** These observations yield two key insights: (1) *Influence of irrelevant passages*: The discrepancy between retrieval recall and RAG accuracy underscores the detrimental effect of irrelevant retrieved passages ("hard negatives") on the LLMs' performance. Even when relevant information is available, the presence of hard negatives can mislead the LLMs and hinder their ability to generate accurate answers. (2) *Limitations of precision as a metric*: The contrasting performance trends observed with e5 and BM25, despite the former's higher precision, reveal that precision alone is an

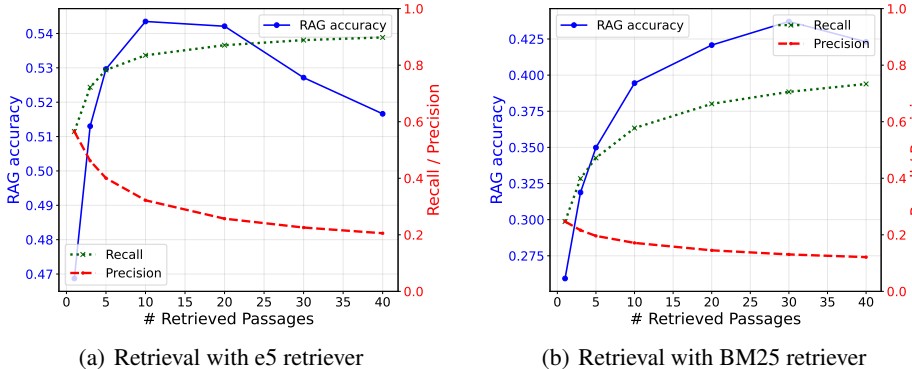

(a) Retrieval with e5 retriever                    (b) Retrieval with BM25 retriever

Figure 2: Analyzing the relationship between RAG performance and retrieval quality (recall/precision) using Gemma-2-9B-Chat with e5 and BM25 retrievers. (1) Accuracy vs. Recall: RAG accuracy consistently falls below retrieval recall for both retrievers, indicating that the presence of relevant information does not guarantee correct answers. This highlights the detrimental impact of irrelevant passages on LLM performance. (2) Precision and hard negatives: Despite higher precision with e5, the performance degradation with increasing retrieval size is more pronounced compared to BM25. This demonstrates that precision alone is an insufficient metric for assessing the impact of "hard negatives," as the nature of irrelevant information significantly influences LLM performance.

inadequate measure of retrieval quality in this context, when the end-to-end performance is considered. The specific characteristics of the irrelevant passages, rather than just their quantity, significantly impact the LLMs' performance. Retrievers might significantly differ in their way of priorization of them, and that might not be fully captured in metrics like precision. In this scenario, it is observed that "hard negatives" retrieved by a stronger retriever (e5) might even more detrimental to the LLM than those retrieved by a weaker one (BM25).

## 3.3 THE IMPORTANCE OF HARD NEGATIVES FOR LONG-CONTEXT LLM EVALUATION

This subsection investigates the impact of "hard negatives" on the performance of long-context LLMs in RAG, highlighting the need for more robust evaluation methodologies.

**Research question.** In long-context RAG scenarios, where a vast knowledge source necessitates retrieving numerous passages, the likelihood of including relevant information (*i.e.* obtaining high recall) increases. However, this also elevates the risk of introducing hard negatives. This raises two critical questions: (1) *How robust are current long-context LLMs to these hard negatives?* and (2) *Does the impact of hard negatives vary with the retriever used?*

**Experimental setting.** This study investigates the effect of hard negative passages on long-context LLM performance in a controlled setting. We tasked three LLMs (Gemma2-7B-Chat, Mistral-Nemo-12B-Instruct, and Gemini-1.5-Pro) with answering queries based on a context comprising a single golden passage and a varying number of hard negative passages retrieved using different methods (e5, Contriever, BM25, and random sampling). This synthetic experiment, detailed in Figure 3, isolates the impact of hard negatives by holding the golden passage constant and intentionally excluding scenarios with multiple golden passages, which are common in real-world RAG systems. See Appendix C for a complete illustration of the experimental setup.

**Observations.** (1) Sensitivity to hard negatives: Across all LLMs, increasing the number of hard negative passages generally leads to a decline in RAG answer accuracy. (2) Retriever strength and hard negative difficulty: The strength of the retriever directly correlates with the difficulty of the retrieved hard negatives. LLMs struggle more with hard negatives from stronger retrievers (*e.g.*, e5) compared to those from weaker retrievers (*e.g.*, BM25) or random sampling. (3) Distinguishing random and hard negatives: While Gemini-1.5-Pro demonstrates robustness to random negatives, it remains susceptible to the influence of hard negatives. More results on other datasets and qualitative studies can be found in Appendix B.2 and D.

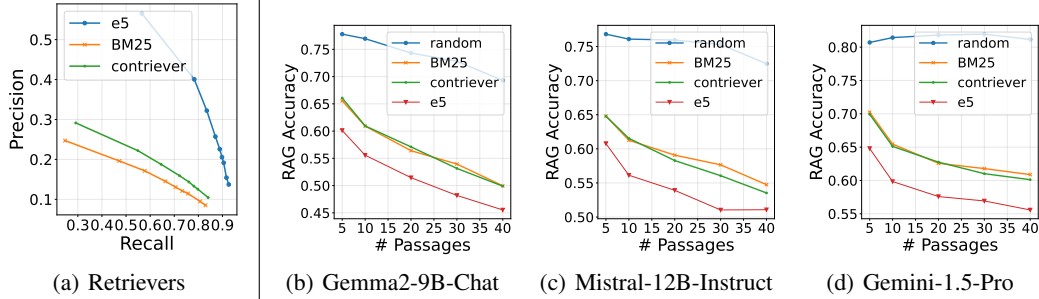

(a) Retrievers   (b) Gemma2-9B-Chat   (c) Mistral-12B-Instruct   (d) Gemini-1.5-Pro

Figure 3: Evaluating the impact of hard negatives on long-context LLMs. (a) The retriever performance on NQ dataset: e5 > contriever > BM25. (b)(c)(d) For each query, a single golden passage (containing the correct answer) is combined with varying numbers of hard negative passages retrieved by different methods: e5, Contriever, BM25, and random sampling. The LLMs are then tasked with answering the query based on this context. This setup allows us to assess the robustness of LLMs to hard negatives and the influence of retriever characteristics on their overall impact.

**Insights.** Existing benchmarks for evaluating long-context LLMs, such as "needle-in-the-haystack" (Kamradt, 2023) and RULER (Hsieh et al., 2024a), predominantly utilize random negatives. Our findings demonstrate that such benchmarks may not adequately capture the challenges posed by hard negatives, which are prevalent in real-world RAG applications. Their takeaways would have limitations. The need for new evaluation methodologies that incorporate hard negatives (specific to the employed retrievers) is highlighted, to provide a more comprehensive and realistic assessment of long-context LLM performance in RAG.

## 4   SIMPLE AND EFFECTIVE TRAINING-FREE RAG IMPROVEMENT

Building upon the analyses in Section 3 on the detrimental impact of hard negatives on long-context LLMs in RAG, we focus on the training-free solution, *retrieval reordering*. This method leverages the inherent "lost-in-the-middle" phenomenon observed in LLMs to mitigate the negative effects of hard negatives. As highlighted by Liu et al. (2024), LLMs exhibit a tendency to prioritize information presented at the beginning and end of an input sequence, while paying less attention to the middle.

Exploiting this "lost-in-the-middle" behavior, we consider a simple and effective strategy: reordering the retrieved passages based on their relevance scores calculated by the retriever. Given a query $q$ and a set of retrieved passages $d_1$, $d_2$, ..., $d_k$ with decreasing relevance scores, the standard input sequence construction for an LLM with instruction $I$ would be: $[I, d_1, d_2, ..., d_{k-1}, d_k, q]$. Retrieval reordering modifies this to prioritize passages with higher scores at the beginning and end: $[I, d_1, d_3, ..., d_4, d_2, q]$ where the position of passage $d_i$ is determined by

$$Order(d_i) = \begin{cases} \frac{i+1}{2} & \text{if } \mod(i, 2) = 1 \\ (k+1) - \frac{i}{2} & \text{if } \mod(i, 2) = 0 \end{cases} \tag{1}$$

This reordering strategy aims to guide the LLM's attention towards the most relevant passages, thereby reducing the influence of hard negatives positioned in the middle of the sequence. The pseudo-code and intuition for retrieval reordering can be found in Appendix E and Q.

**Retrieval reordering significantly improves RAG performance, particularly with larger numbers of retrieved passages.** To assess the effectiveness of retrieval reordering, we conduct experiments with two retrievers (e5 and BM25), two long-context LLMs (Gemma-2-9B-Chat and Mistral-Nemo-12B-Instruct), and two datasets (NQ and PopQA). As illustrated in Figure 4, retrieval reordering yields negligible improvements with smaller retrieval sets, but significantly and consistently outperforms the original ordering when the number of retrieved passages is large. This behavior is attributed to the interplay of two factors that become increasingly significant with larger retrieval sets: (1) the amplified "lost-in-the-middle" phenomenon, where LLMs prioritize information at the beginning and end of the input sequence, and (2) the increased prevalence of hard negatives, which can hinder accurate answer generation. By strategically placing passages, retrieval reordering mitigates these issues, highlighting the potential of *position engineering* as a complementary technique to prompt engineering for optimizing long-context LLMs in RAG.

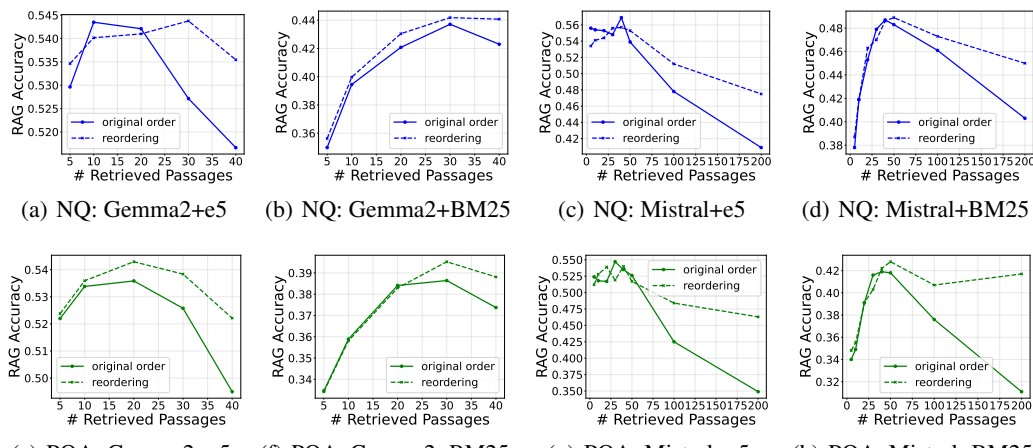

Figure 4: Evaluating the effectiveness of retrieval reordering in various RAG configurations. Results demonstrate that reordering retrieved passages consistently enhances performance, particularly when the number of retrieved passages is large. (Retrievers: e5, BM25; LLMs: Gemma2-9b-Chat, Mistral-Nemo-12B-Instruct; Datasets: NQ, PopQA)

# 5 IMPROVING ROBUSTNESS FOR RAG VIA DATA-AUGMENTED FINE-TUNING

## 5.1 IMPLICITLY IMPROVING LLM ROBUSTNESS THROUGH FINE-TUNING

While the *retrieval reordering* strategy presented in Section 4 mitigates the detrimental impact of hard negatives, it does not inherently enhance the LLM's ability to handle such irrelevant information within the context. To address this, we conduct a systematic investigation into RAG-specific tuning as a means of improving long-context LLMs for RAG applications.

Our tuning paradigm involves training LLM to generate the correct answer ($a$) given a comprehensive input comprising an instruction ($I$), a query ($q$), and a set of retrieved passages ($d_1, d_2, ..., d_k$):

$$\text{Input: } [I, d_1, d_2, ..., d_{k-1}, d_k, q] \longrightarrow \text{Output: } a. \tag{2}$$

This approach aims to implicitly enhance the LLM's robustness to hard negatives by exposing it to a diverse range of retrieved contexts during fine-tuning, thus enabling it to learn to effectively identify and utilize relevant information even in the presence of noise.

To assess the generalization capabilities of RAG-tuned LLMs, we fine-tune Gemma-2-9B-Base, Mistral-Nemo-12B-Base and Gemini-1.0-Pro using a diverse dataset comprising NQ, WoW, Fever, and MMLU. We then evaluate on a range of unseen datasets, including TriviaQA, PopQA, HotpotQA, 2wikimultihopqa, Webquestions, Bamboogle, ASQA, T-REx, and zsRE. We use e5 as the retriever and Wikipedia-18 (Karpukhin et al., 2020) as the corpus. We compare the performance of the RAG-tuned model (RAG FT) with two types of baselines: (1) Chat model with retrieval augmentation: the Gemma-2-9B-Chat/Mistral-Nemo-12B-Instruct/Gemini-1.0-Pro w. RAG; (2) Direct SFT: the ones fine-tuned with standard supervised fine-tuning (SFT) on question-answer pairs without retrieved context (Direct FT w/o RAG). Further details regarding the datasets and experimental setup can be found in Appendix F and G.

Figure 5 shows the three key observations: (1) Consistent improvement over baselines: RAG FT consistently outperforms the chat model w. RAG and the Direct FT model across all evaluated datasets. (2) Robustness to hard negatives: the curve of RAG FT is generally flatter than that of the chat model, which demonstrates that our finetuned LLM is more robust to the hard negatives as the number of retrieved passages increases. (3) Superiority over direct fine-tuning: In most cases, RAG FT demonstrates superior performance compared to Direct FT. This indicates that RAG FT not only enables the LLM to "memorize" knowledge during training but also equips it with the ability to effectively "extract" relevant information from retrieved context during inference. These findings highlight the effectiveness of RAG-specific tuning in enhancing the generalization capabilities of LLMs for knowledge-intensive tasks. Separate results on those three LLMs are shown in Appendix J, K and L. Qualitative studies can be found in Appendix I.

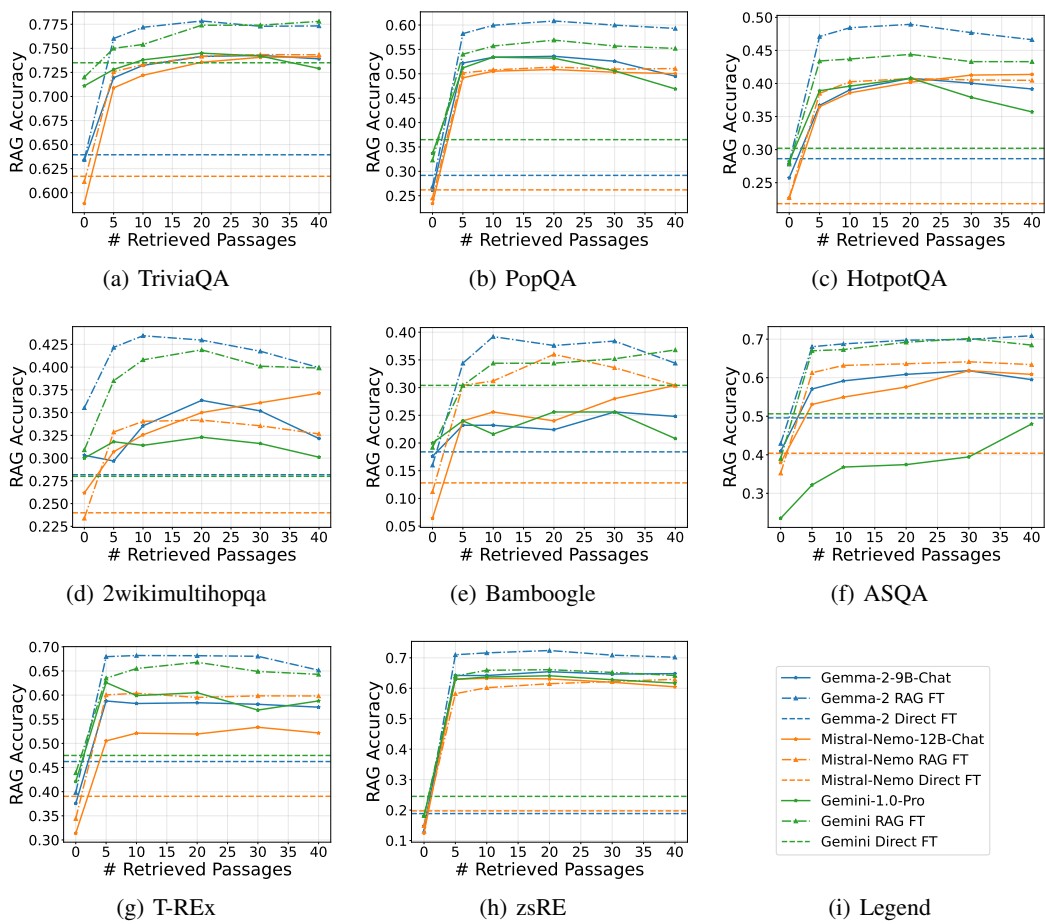

(a) TriviaQA     (b) PopQA     (c) HotpotQA

(d) 2wikimultihopqa     (e) Bamboogle     (f) ASQA

(g) T-REx     (h) zsRE     (i) Legend

Figure 5: Generalization ability of LLMs fine-tuned with RAG-specific data (RAG FT). RAG FT consistently outperforms the chat LLM w. RAG and the model fine-tuned directly on question-answer pairs (Direct FT). This demonstrates the effectiveness of RAG FT in enabling the LLM to effectively extract knowledge from retrieved context on unseen tasks. Note that Direct FT is evaluated without retrieval to align with its training paradigm and all others are evaluated with retrieval augmentation. (LLMs: Gemma-2-9B-Base, Mistral-Nemo-12B-Base, Gemini-1.0-Pro)

## 5.2 ENHANCING RELEVANCE IDENTIFICATION THROUGH REASONING AUGMENTATION

While the fine-tuning approach described in Section 5.1 implicitly enhances the LLM's robustness to hard negatives, it does not explicitly train the model to differentiate between relevant and irrelevant passages within the retrieved context. To address this, we investigate the effectiveness of incorporating an intermediate reasoning step into the fine-tuning process.

This modified paradigm involves training the LLM to generate both a reasoning paragraph ($r$) that explicitly identifies the relevant passages for the given query ($q$) and the final answer ($a$):

$$\text{Input: } [I, d_1, d_2, ..., d_{k-1}, d_k, q] \longrightarrow \text{Output: } [r, a], \tag{3}$$

During training, the LLMs are provided with labeled reasoning paragraphs to guide its learning process. During inference, the LLMs are instructed to first generate the reasoning paragraph and then utilize this analysis to produce the answer. This approach aims to explicitly enhance the LLMs' ability to discern relevant information from noise within the retrieved context, thereby improving its overall performance in RAG.

We utilize the same training data mixture as in Section 5.1 and augment it with reasoning labels generated by Gemini-1.5-Pro for each question-passage pair. These labels provide explicit guidance on identifying relevant passages. We use e5 as the retriever and Wiki-18 as the corpus. Further details of the experimental setup and the generation of reasoning labels can be found in Appendix H.

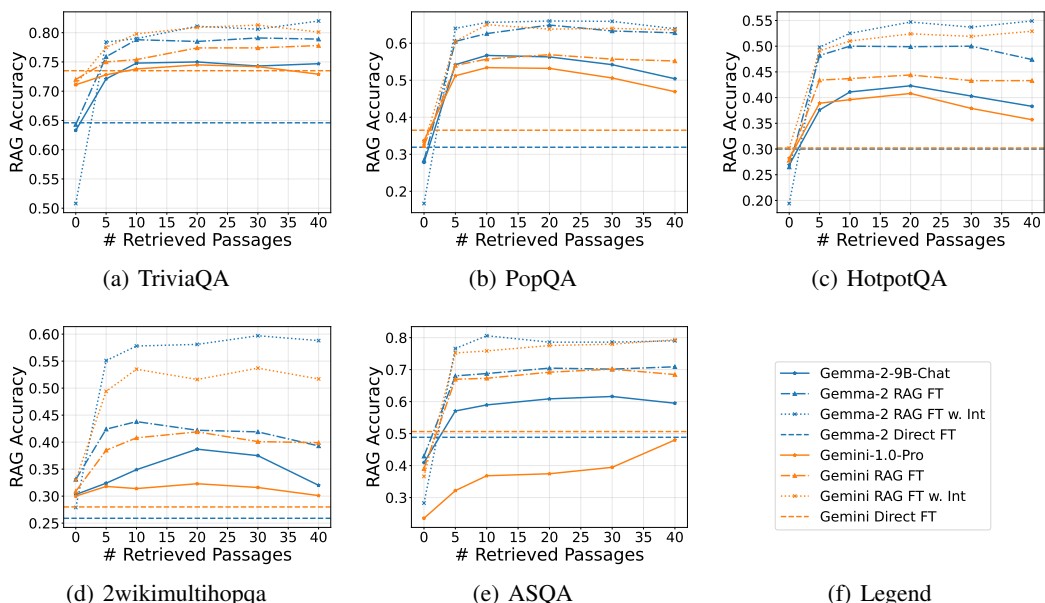

(a) TriviaQA      (b) PopQA      (c) HotpotQA

(d) 2wikimultihopqa      (e) ASQA      (f) Legend

Figure 6: Evaluating the impact of intermediate reasoning on the performance of RAG-tuned LLMs. Results demonstrate that fine-tuning with an intermediate reasoning step (RAG FT w. Int) leads to further improvements compared to implicit RAG fine-tuning (RAG FT) and direct fine-tuning (Direct FT). Direct FT is evaluated without retrieval to align with its training and all others are evaluated with retrieval augmentation. Due to the computational complexity of inference with reasoning augmentation, results are shown for 1000 randomly-sampled queries from each dataset. (LLMs: Gemma-2-9B-Base and Gemini-1.0-Pro, more results in Appendix J, K and L)

Figure 6 demonstrates the effectiveness of this approach. The LLM fine-tuned with explicit intermediate reasoning consistently outperforms training with implicit RAG data. This improvement can be attributed to two key factors: (1) Explicit relevance training: Providing intermediate reasoning labels during training explicitly teaches the LLM to differentiate between relevant and irrelevant passages, enhancing its ability to discern crucial information from noise. (2) Structured reasoning for enhanced understanding: Generating a reasoning paragraph before answering introduces a structured approach to processing the retrieved context. This step, akin to chain-of-thought reasoning (Wei et al., 2022), helps decouple the complex information and facilitates a more focused analysis, ultimately leading to improved performance. These highlight the value of incorporating explicit reasoning mechanisms in RAG tuning to enhance the LLM's ability to effectively utilize retrieved context. More results on Gemma-2-9B models, Mistral-Nemo-12B models and Gemini-1.0-Pro models are shown in Appendix J, K and L. Qualitative studies and cost-benefit analysis can be found in Appendix I and P.

## 6 DATA-CENTRIC PERSPECTIVES ON FINE-TUNING LLMS FOR RAG

**Impact of training data distribution on generalization.** We first examine how the distribution of training data affects the generalization of the fine-tuned LLM. We train LLMs on five different data distributions, each with 50k samples: (1) a mixed dataset comprising NQ, WoW, Fever, and MMLU (12.5k samples from each); (2) NQ only; (3) WoW only; (4) Fever only; and (5) MMLU only.

Figure 7(a) demonstrates that a mixed distribution of training data leads to superior generalization performance on unseen RAG tasks compared to training on a single data source. This highlights the importance of data diversity in enhancing the adaptability of LLMs to new RAG scenarios.

**Influence of retrievers on generalization.** In real-world RAG deployments, LLMs might be paired with different retrievers depending on specific external knowledge corpus and retrievers' capabilities. To investigate the impact of different retrievers on fine-tuning, we explore three adaptation scenarios on NQ: fine-tuning with (1) passages retrieved by BM25 (FT w. BM25); (2) passages retrieved by e5 (FT w. e5); and (3) mixture of passages retrieved by both BM25 and e5 (FT w. mix). We evaluate the performance of these fine-tuned LLMs using both retrievers seen during training (BM25 and e5) and unseen retrievers (Contriever (Izacard et al., 2021) and BGE (Chen et al., 2024a)).

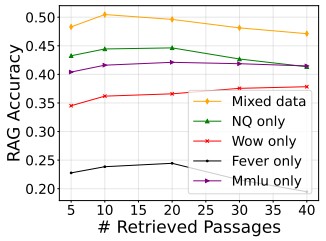
(a) Analysis of training data distribution. (Test: HotpotQA)

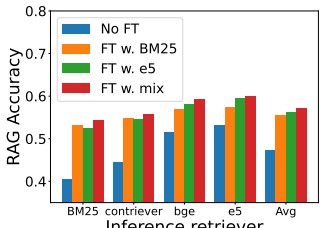
(b) Influence of retriever variations on fine-tuning effectiveness. (NQ)

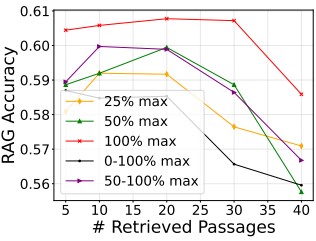
(c) Investigation of the optimal number of passages for training.

Figure 7: (a) Impact of training data distribution: A diverse mix of training data sources enhances the generalization ability of the LLM. (b) Influence of the retriever choice: Fine-tuning with data retrieved from multiple retrievers improves generalization to unseen retrievers during inference. (c) Effect of training context length: Fine-tuning with the maximum context length yields optimal performance across varying numbers of retrieved passages during inference. (LLM: Gemma-2-9B-Base)

Figure 7(b) presents the results, revealing three key findings: (1) Superiority of mixed retriever training: Fine-tuning with the data corresponding to a mix of retrievers consistently yields the best performance across both seen and unseen retrievers during inference. This suggests that training on a diverse set of retrieved passages enhances the LLMs' ability to adapt to different retrieval strategies and knowledge sources. (2) Retriever-agnostic improvement: Our proposed RAG-specific tuning consistently enhances LLM performance across all retrievers, demonstrating the generalizability of our approach. (3) Retriever similarity and generalization: The generalization ability of an LLM fine-tuned with a specific retriever is influenced by the similarity between the training retriever and the inference retriever. For instance, an LLM trained with BM25 generalizes better to Contriever, while an LLM trained with e5 generalizes better to BGE. This observation suggests that "hard negatives" exhibit different characteristics depending on the employed retriever, and training with a specific retriever implicitly equips the LLM to better handle similar types of hard negatives. See Appendix A for a detailed analysis of retriever similarity.

**Optimizing training for variable retrieval sizes.** In real-world RAG systems, the number of retrieved passages can vary depending on the specific knowledge source and user requirements. Therefore, it is essential to determine the optimal training strategy for LLMs to ensure robust performance across different retrieval sizes during inference. We investigate this aspect with the Gemma-2-9B-Base model, which has a maximum input sequence length of 8192 tokens (corresponding to approximately 40 passages). We evaluate five different training configurations: (1) Fixed 10 retrieved passages (25% max). (2) Fixed 20 retrieved passages (50% max). (3) Fixed 40 retrieved passages (maximum input capacity) (100% max). (4) Dynamic 0-40 retrieved passages (0-100% max). (5) Dynamic 20-40 retrieved passages (50-100% max).

Figure 7(c) presents the results on NQ, demonstrating that fine-tuning with the maximum number of retrieved passages (100% max) consistently yields the best performance across various retrieval sizes during inference. This suggests that training with the full context capacity enhances the LLM's ability to effectively handle varying amounts of retrieved information, leading to improved generalization and robustness. More analyses of RAG-specific tuning can be found in in Appendix M and N.

# 7 CONCLUSIONS

This paper investigates the impact of increasing the number of retrieved passages on the performance of long-context LLMs in retrieval-augmented generation (RAG) systems. Contrary to expectations, we observe that performance initially improve but then degrade as more passages are included. This phenomenon is attributed to the detrimental influence of retrieved "hard negatives". To mitigate this issue, we propose and evaluate three solutions: training-free retrieval reordering, RAG-specific implicit LLM fine-tuning, and RAG-oriented LLM fine-tuning with intermediate reasoning. A systematic analysis of the training-based methods explores the effects of data distribution, retriever for training, and training context length. Interesting future directions include exploring (automated) position optimization with more advanced retrieval ordering methods, and fine-tuning the LLMs for RAG with more fine-grained and multi-step reasoning chains. More discussion on the limitation and future works can be found in Appendix S.

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

APPENDIX

## A    RETRIEVER PERFORMANCE AND SIMILARITY

We analyze the performance and similarity of four retrievers (BM25, contriever, e5 and bge) on the NQ dataset shown in Figure 8. Each data point corresponds to a retrieval (recall, precision) pair for a specific number of retrieved passages. The overall retrieval performances on NQ are observed as e5 > bge > contriever > bm25, with contriever having a similar performance with BM25 and bge having a similar performance with e5 (as their curves are closer).

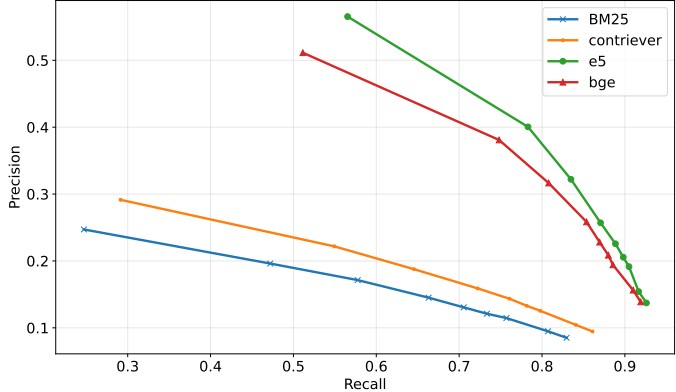

Figure 8: Retriever performance on NQ. (1) Retrieval performance: e5 > bge > contriever > BM25; (2) Contriever is more similar to BM25, while bge is more similar to e5 (since their curves are closer respectively).

## B    LONG CONTEXT LLMS IN RAG ANALYSIS ON OTHER DATASETS

In addition to the analysis presented on the NQ dataset in Section 3, we conduct further studies on the PopQA dataset to underscore the generality of our findings.

### B.1    THE EFFECT OF RETRIEVED CONTEXT SIZE ON RAG PERFORMANCE

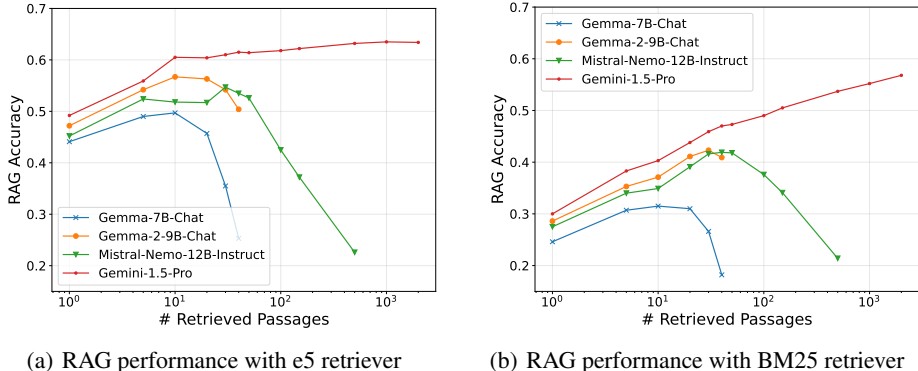

(a) RAG performance with e5 retriever          (b) RAG performance with BM25 retriever

Figure 9: Impact of retrieved context size on RAG performance (on PopQA) with 4 different LLMs. Increasing the number of retrieved passages initially improves performance but then leads to a decline. This degradation is more pronounced using a retriever (e5) that exhibits higher recall@k on PopQA compared to BM25 (Recall@40 is 0.85 with e5 and 0.57 with BM25). The maximum number of retrieved passages varies across LLMs due to differences in their maximum token limits.

**Observations.** Figure 9 presents the following key observations similar to that in Section 3.1: 1) Strong Retriever (e5): Across all LLMs, increasing the number of retrieved passages initially enhances performance, but subsequently results in either a sharp decline or a plateau. 2) Weak Retriever (BM25): Performance generally shows a continuous improvement or a slighter decrease as the number of retrieved passages increases. While these observations may appear counter-intuitive - given that one might expect monotonic improvements due to higher recall (*i.e.*, a greater chance of retrieving relevant information) - the inclusion of additional passages can reduce precision, with irrelevant or misleading passages detracting LLMs from overall performance.

## B.2 THE IMPORTANCE OF HARD NEGATIVES FOR LONG-CONTEXT LLM EVALUATION

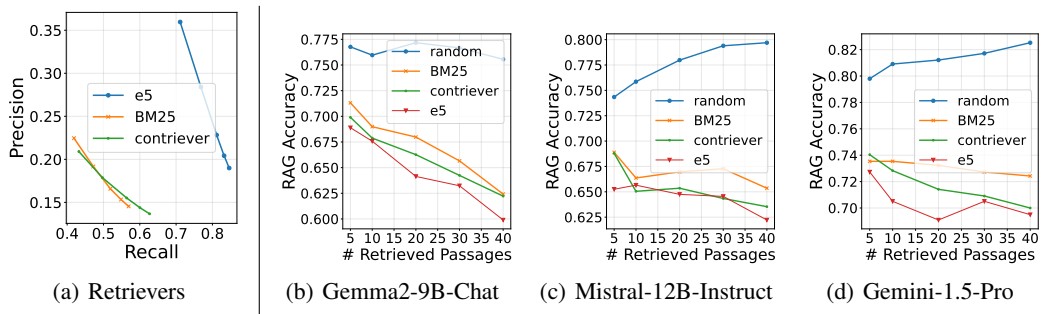

(a) Retrievers     (b) Gemma2-9B-Chat     (c) Mistral-12B-Instruct     (d) Gemini-1.5-Pro

Figure 10: Evaluating the impact of hard negatives on long-context LLMs. (a) The retriever performance on PopQA dataset: e5 > contriever > BM25. (b)(c)(d) For each query, a single golden passage (containing the correct answer) is combined with varying numbers of hard negative passages retrieved by different methods (e5, Contriever, BM25, and random sampling). The LLMs are then tasked with answering the query based on this context. This setup allows us to assess the robustness of LLMs to hard negatives and the influence of retriever strength on their impact.

**Observations.** Figure 10 shows the following observations similar to that in Section 3.3: (1) Sensitivity to hard negatives: Across all LLMs, increasing the number of hard negative passages generally results in a decline in RAG answer accuracy. (2) Retriever strength and hard negative difficulty: The strength of the retriever is directly correlated with the difficulty of the retrieved hard negatives. LLMs struggle more with hard negatives generated by stronger retrievers (*e.g.*, e5) compared to those produced by weaker retrievers (*e.g.*, BM25) or through random sampling. (3) Distinguishing random and hard negatives: While all the LLMs demonstrates robustness to random negatives, it remains susceptible to the influence of hard negatives.

## C    ILLUSTRATION OF SECTION 3.3: HARD NEGATIVE STUDY

---

**Algorithm 1** Data Construction for Hard Negative Study

---

**Require:** Query $q$, instruction $I$, golden passage $d_{\text{gold}}$, golden answer $a$, retrieved passages $D = [d_1, d_2, \ldots, d_N]$ with decreasing retriever relevance scores, desired number of passages $K$ ($K \ll N$).

**Ensure:** Input sequence $S$.

 1: Initialize list $S \leftarrow [d_{\text{gold}}]$
 2: **for** each passage $d_i$ in $D$ **do**
 3:    **if** $d_i \neq d_{\text{gold}}$ **and** $a$ not in $d_i$ **then**
 4:       Append $d_i$ to $S$
 5:    **end if**
 6:    **if** $|S| = K$ **then**
 7:       **break**
 8:    **end if**
 9: **end for**
10: Randomly shuffle $S$.
11: Construct input sequence $[I, S[1], S[2], \ldots, S[K], q]$.
12: **return**  The input sequence $[I, S[1], S[2], \ldots, S[K], q]$.

---

## D    HARD NEGATIVES CASE STUDY

In this section, we provide a case study to compare the hard negatives returned by different retrievers. We classify the negative passages into two types: (1) *Related but Irrelevant*: passages related to some entities mentioned in the question but not containing the ground truth answer; (2) *Not Related*: passages not related to the question at all. Note that *Related but Irrelevant* passages are harder and more misleading to the LLMs compared with *Not Related* passages. We show the top-5 negatives from each retriever for a random sampled question as below:

| **Question** | The south west wind blows across Nigeria between? |
|---|---|
| **Ground Truth** | Till September |

| **Retrieved Hard Negative Passages** (high retrieval score but lacking ground truth answer) | |
|---|---|
| **w. e5** | **Doc 1** (Title: "Geography of Nigeria") ... south atlantic ocean, locally known as the south western wind, or by its main name, The Tropical Maritime (MT) airmass. These two major wind systems in Nigeria are known as the trade winds. The tropical maritime airmass (MT) is responsible for Nigeria's rainy season. This wind (the tropical maritime airmass) invades the country from February in the southern part of Nigeria while it takes longer for the wind to fully cover the whole of the country, reaching the northern part of Nigeria in June. Its invasion is as a result of the northward retreat, ... *[Related but Irrelevant]*

**Doc 2** (Title: "Onikwu") ... The dry season is accompanied by a dust laden airmass from the Sahara Desert, locally known as Harmattan, or by its main name, The Tropical Continental (CT) airmass, while the rainy season is heavily influenced by an airmass originating from the south Atlantic Ocean, locally known as the south west wind, or by its main name, The Tropical Maritime (MT) airmass. These two major wind systems in Nigeria are known as the trade winds. The region Onikwu/Ndoni is flood prone communities, this is because the inland part of Rivers state consists of tropical rainforest ... *[Related but Irrelevant]*

**Doc 3** (Title: "Geography of Nigeria") ... northern end is south of the 15 degrees line at about 14 degrees. Nigeria's location in the wetter part of the easterly waves south of the 15 degree line creates wetter climatic conditions for Nigeria especially during the monsoons. The Tropical Continental Airmass (CT) locally known as the harmattan, is a wind originating from North Africa which crosses the Sahara Desert into west Africa to Nigeria. This airmass dominates Nigeria's climate during the dry season from December to March. The Tropical continental airmass is dusty and creates a haze within the atmosphere of west Africa and Nigeria when it predominates. ... *[Related but Irrelevant]*

**Doc 4** (Title: "Nigeria") ... Niger, Chad, Cameroon, and has a coastline of at least s. Nigeria lies between latitudes 4 and 14N, and longitudes 2 and 15E. The highest point in Nigeria is Chappal Waddi at . The main rivers are the Niger and the Benue, which converge and empty into the Niger Delta. This is one of the world's largest river deltas, and the location of a large area of Central African mangroves. Nigeria has a varied landscape. The far south is defined by its tropical rainforest climate, where annual rainfall is a year. In the southeast stands the ... *[Related but Irrelevant]*

**Doc 5** (Title: "Geography of Nigeria") ... Nigeria, has a temperature range of to , and an annual rainfall of about with a single rainfall maxima in September. The single Dry season experienced in this climate, the tropical savanna climate in central Nigeria beginning from December to march, is hot and dry with the Harmattan wind, a continental tropical (CT) airmass laden with dust from the Sahara Desert prevailing throughout this period. With the Intertropical Convergence Zone (ITCZ) swinging northward over West Africa from the Southern Hemisphere in April, heavy showers coming from pre-monsoonal convective clouds mainly in the form of ... *[Related but Irrelevant]* |

| | |
|---|---|
| **w. bm25** | **Doc 1** (Title: "Oron people") ... Civil War. Oron is found in the flood plain of South Eastern Nigeria, with the land mainly intersected by numerous streams and tributaries flowing into Cross River. The entire coastline stretches from Uya Oron to Udung Uko. Oron is in the tropical region and has a uniformly high temperature all the year round. The two main seasons are the dry which spans between October and April and wet season which starts around May and ends in September. There are also two prevailing winds – the South-West onshore winds which brings heavy rains and the ... *[Not Related]*

**Doc 2** (Title: "South Equatorial Current") ... is driven directly by the trade winds which blow from east to west. In the Indian Ocean, the westward-flowing South Equatorial Current is well-developed only south of the equator. Directly on the equator, the winds reverse twice a year due to the monsoons, and so the surface current can be either eastward or westward. South Equatorial Current Ocean current in the Pacific, Atlantic, and Indian Ocean that flows east-to-west between the equator and about 20 degrees south. In the Pacific and Atlantic Oceans, it extends across the equator to about 5 degrees north. Within the southern hemisphere, the South Equatorial ... *[Related but Irrelevant]*

**Doc 3** (Title: "Wind direction") ... Wind direction Wind direction is reported by the direction from which it originates. For example, a ""northerly"" wind blows from the north to the south. Wind direction is usually reported in cardinal directions or in azimuth degrees. Wind direction is measured in degrees clockwise from due north. Consequently, a wind blowing from the north has a wind direction of 0; a wind blowing from the east has a wind direction of 90; a wind blowing from the south has a wind direction of 180; and a wind blowing from the west has a wind direction of 270 ... *[Related but Irrelevant]*

**Doc 4** (Title: "Gulf Stream") ... this current interacts with the northeastern coast of South America, the current forks into two branches. One passes into the Caribbean Sea, while a second, the Antilles Current, flows north and east of the West Indies. These two branches rejoin north of the Straits of Florida. The trade winds blow westward in the tropics, and the westerlies blow eastward at mid-latitudes. This wind pattern applies a stress to the subtropical ocean surface with negative curl across the north Atlantic Ocean. The resulting Sverdrup transport is equatorward. Because of conservation of potential vorticity caused by the northward-moving winds on the subtropical ... *[Not Related]*

**Doc 5** (Title: "Climate of the United Kingdom") ... climate that western parts of the UK experience. The high latitude and proximity to a large ocean to the west means that the United Kingdom experiences strong winds. The prevailing wind is from the south-west, but it may blow from any direction for sustained periods of time. Winds are strongest near westerly facing coasts and exposed headlands. Gales — which are defined as winds with speeds of — are strongly associated with the passage of deep depressions across the country. The Hebrides experience on average 35 days of gale a year (a day where there are gale-force winds) while inland ... *[Not Related]* |

| | |
|---|---|
| **w. random sampling** | **Doc 1** (Title: "Queen of Peace, Bray") ... of Bugisi in Tanzania for a number of years. The parish has a very close relationship with St Cronan's B.N.S., Scoil Chualann and Gaelscoil Uí Chéadaigh. The parish provides the sacraments of Communion and Confirmation to the children in the schools. They also help to raise funds for the twin parish of Bugisi. Queen of Peace, Bray The Queen of Peace is a Catholic church situated at the junction of the Putland Road and the Vevay Road in Bray, Co. Wicklow, Ireland. The present church was built in 1946 by TJ Macken of St Patrick's Street, Dún Laoghaire, ... *[Not Related]*

**Doc 2** (Title: "Cordova Congressional Internship Program") ... Puerto Rico's Constitutional Convention from 1951 to 1952. By 2012, over 670 students from colleges and universities in Puerto Rico had enjoyed internships under the program, and the Spring 2009 class included a record 24 members. A private sector committee, recently headed by Univision Puerto Rico president Larry Sands, provides private funds to supplement the 350,000 annual grant provided by the Puerto Rico Legislative Assembly. Under the auspices of TWC, seventeen states have since established similar legislative-funded Congressional internship programs. The Center established in 2008 the McClintock Award to the State Legislator of the Year ... *[Not Related]*

**Doc 3** (Title: "V bomber") ... Puerto Rico's Constitutional Convention from 1951 to 1952. By 2012, over 670 students from colleges and universities in Puerto Rico had enjoyed internships under the program, and the Spring 2009 class included a record 24 members. A private sector committee, recently headed by Univision Puerto Rico president Larry Sands, provides private funds to supplement the 350,000 annual grant provided by the Puerto Rico Legislative Assembly. Under the auspices of TWC, seventeen states have since established similar legislative-funded Congressional internship programs. The Center established in 2008 the McClintock Award to the ... *[Not Related]*

**Doc 4** (Title: "Defence Materials and Stores Research and Development Establishment") ... materials for the Indian Armed Forces. DMSRDE has developed Nuclear Shielding Pad, Boot Anti Mine, Blast Protection Suit, Bullet Proof Jackets, etc.. ""The Defence Material and Stores Research Development Establishment in Kanpur has developed a new NBC suit that would be proved effective against any kind of dangerous weapons or chemicals and protect soldiers from any sort of attack,"" DMSRDE Director Arvind Kumar Saxena was quoted by media-persons. 40,000 pieces of NBC suits costing about Rs 30,000 had been requested by Indian army. ""the further progress on the other two suits are going on."" further ... *[Not Related]*

**Doc 5** (Title: "Chess title") ... retain the title of Candidate Master, if it was earned according to criteria above). This is in contrast to international titles awarded by FIDE, which are awarded for life. In European countries the term of ""expert"" is not used. Instead, players of that level are called ""Candidate Masters"", although the FIDE Candidate Master title generally requires a higher rating (2200 FIDE). It is possible (and common), however, for players in the United States to have a rating that places them in the 'expert' category while still retaining the title of 'Life Master' or 'National Master' ... *[Not Related]* |

From the case study, we can find that the negatives retrieved by e5 contain more *Related but Irrelevant* passages compared with those retrieved by bm25, while those retrieved by bm25 still have more *Related but Irrelevant* passages than random sampling. This qualitatively demonstrates that the hardness of the negatives from different retrievers as e5 > bm25 > random.

# E  RETRIEVAL REORDERING

---

**Algorithm 2** Retrieval Reordering Algorithm

---

**Require:** Query $q$, instruction $I$, retrieved passages $D = [d_1, d_2, ..., d_k]$ with decreasing retriever relevance scores.

**Ensure:** Reordered sequence $S$.

1: Initialize an empty list $S$ of length $k$.
2: **for** $i = 1$ to $k$ **do**
3:     **if** $\mathrm{mod}(i, 2) = 1$ **then**
4:         $Order(d_i) \leftarrow \dfrac{i+1}{2}$ $\{i$ is odd$\}$
5:     **else**
6:         $Order(d_i) \leftarrow k + 1 - \dfrac{i}{2}$ $\{i$ is even$\}$
7:     **end if**
8:     Place $d_i$ at position $Order(d_i)$ in $S$.
9: **end for**
10: Construct input sequence $[I, S[1], S[2], ..., S[k], q]$.
11: **return** The reordered sequence $[I, S[1], S[2], ..., S[k], q]$.

---

# F  DATASETS

In this section, we discuss the datasets for RAG-specific LLM training and evaluation.

## F.1  TRAINING DATASETS

| Dataset | the number of instances |
|---|---|
| Natural Question | 12,500 |
| Wizard of Wikipedia | 12,500 |
| FEVER | 12,500 |
| MMLU | 12,500 |

Table 2: Training data statistics.

We select a series of fine-tuning data designed to enhance the model's robustness to hard negatives in the retrieval context and improve its contextual awareness in generating predictions. The training data are from four sources with different answer types: Natural Question (short-form), Wizard of Wikipedia (long-form), FEVER (true/false), and MMLU (close-set). The statistics of the training data mix can be found in Table 2.

## F.2  TESTING DATASETS

To comprehensively evaluate our methods, we select testing datasets across different tasks including: (1) Question-answering: TriviaQA, PopQA, WebQuestions; (2) Multi-hop tasks: HotpotQA, 2Wiki-MultiHopQA, Bamboogle; (3) Long-form tasks: ASQA; (4) Slot filling: T-REx, Zero-shot RE. The statistics of all the datasets can be found in Table 3.

## F.3  RETRIEVAL CORPUS

Following Karpukhin et al. (2020), we use the text chunks from 2018 Wikipedia dump as the retrieval corpus. The articles are split by section, where long sections are further split into text chunks of equal sizes and contain less than 100 words, leading to a total of 21M text chunks.

| Dataset | Task | the number of instances |
|---|---|---|
| TriviaQA | QA | 11,313 |
| PopQA | QA | 14,267 |
| WebQuestions | QA | 2,032 |
| HotpotQA | Multi-Hop QA | 7,405 |
| 2WikiMultiHopQA | Multi-Hop QA | 12,576 |
| Bamboogle | Multi-Hop QA | 125 |
| ASQA | Long-form QA | 948 |
| T-REx | Slot filling | 5,000 |
| Zero-shot RE | Slot filling | 3,724 |

Table 3: Testing data statistics.

## G  IMPLICIT RAG FINE-TUNING EXPERIMENTAL SETTING

### G.1  TRAINING SETTINGS

**Hyperparameters.** We use the top-40 retrieved text chunks for a given example to generate the fine-tuning samples and use e5 as the retriever for the main results. We fine-tune both Gemma-2-9B-Base and Mistral-Nemo-12B-Base using 8 H100 GPUs. For both models, we use the chat template corresponding to Gemma-2-9B-Chat and Mistral-Nemo-12B-Instruct respectively when tuning the models. We use the axolotl[§] codebase for their tuning. For Gemini-1.0-Pro tuning, we use the Google Cloud Tuning API[§] with the default settings. The hyperparameters can be found in Table 4.

| Model | peak lr | lr scheduler | warm up | # epoch | batch size | Flash Att |
|---|---|---|---|---|---|---|
| Gemma-2-9B-Base | 1e-6 | cosine | 5% | 4 | 64 | False |
| Mistral-Nemo-12B-Base | 1e-6 | cosine | 5% | 4 | 64 | True |
| Gemini-1.0-Pro | default | default | default | 1 | default | default |

Table 4: Implicit RAG finetuning hyperparameters.

**Training RAG instruction templates.** The RAG instruction templates for different training datasets can be found in Table 5.

| Task | Instruction Templates |
|---|---|
| NQ | Answer the question based on the given document. Only give me the answer and do not output any other words. The following are given documents. {reference} |
| Wizard of Wikipedia | Provide a response to the conversation based on the given document. The following are given documents. {reference} |
| FEVER | Verify a fact based on the given documents The following are given documents. {reference} |
| MMLU | Given a question, choose the answer from the options based on the given documents. The following are given documents. {reference} |

Table 5: Training instruction templates for implicit RAG tuning.

---

[§] https://github.com/axolotl-ai-cloud/axolotl
[§] https://cloud.google.com/vertex-ai/generative-ai/docs/model-reference/tuning

**Training RAG answer templates.** The RAG answer templates for different training datasets can be found in Table 6.

| Task | Answer Templates |
|------|------------------|
| NQ | Question: {question}
Answer: |
| Wizard of Wikipedia | Conversation: {question}
Response: |
| FEVER | Return SUPPORTS if it is correct and return REFUTES if it is not correct.
Fact: {question}
Response: |
| MMLU | Question: {question}
Options: {choices}
Answer: |

Table 6: Training answer templates for implicit RAG tuning.

## G.2 EVALUATION SETTINGS

**Hyperparameters.** For all the compared LLMs, we conduct top-p sampling (p = 1) and the maximum number of generated token is set to be 32. For Gemma-2 series models, we use the huggingface inference pipeline[§]. For Gemini series models, we use Google Cloud Inference API[§]. While for other series of LLMs, we utilize vLLM[§] codebase for efficient generation.

**Evaluation RAG instruction templates.** The RAG instruction templates for different testing datasets can be found in Table 7.

| Task | Instruction Templates |
|------|-----------------------|
| QA | Answer the question based on the given document.
Only give me the answer and do not output any other words.
The following are given documents.
{reference} |
| Multi-hop | Answer the question based on the given document.
Only give me the answer and do not output any other words.
The following are given documents.
{reference} |
| Long-form | Answer the question based on the given document.
Please give in-depth explanation and avoid only returning the answer.
The following are given documents.
{reference} |
| Slot filling | Given a question, choose the answer from the options based on the given documents.
The following are given documents.
{reference} |

Table 7: Testing instruction templates for implicit RAG tuning.

**Evaluation RAG answer templates.** The RAG answer templates for different testing datasets are all: "Question: {question}. Answer:"

---

[§]https://huggingface.co/docs/transformers/
[§]https://cloud.google.com/vertex-ai/generative-ai/docs/model-reference/inference
[§]https://github.com/vllm-project/vllm

# H   RAG FINETUNING WITH INTERMEDIATE REASONING EXPERIMENTAL SETTING

## H.1   TRAINING SETTINGS

**Hyperparameters.**   The hyperparameter setting is the same to that in Appendix G.1.

**Training RAG instruction templates.**   The RAG instruction templates with intermediate reasoning for different training datasets can be found in Table 8.

| Task | Instruction Templates |
|------|------------------------|
| NQ | Answer the question based on the given documents. Please first provide an analysis with clear reasoning details of which documents are relevant to answer the question. Then output a concise answer to the question based on the analysis. The following are given documents. {reference} |
| Wizard of Wikipedia | Provide a response to the conversation based on the given documents. Please first provide an analysis with clear reasoning details of which documents are relevant to provide the response. Then output a concise response to the question based on the analysis. The following are given documents. {reference} |
| FEVER | Verify a fact based on the given documents. Please first provide an analysis with clear reasoning details of which documents are relevant to verify the fact. Then output a concise answer (SUPPORTS or REFUTES) based on the analysis. The following are given documents. {reference} |
| MMLU | Given a question, choose the answer from the options based on the given documents. Please first provide an analysis with clear reasoning details of which documents are relevant to answer the question Then output a concise answer to the question based on the analysis. The following are given documents. {reference} |

Table 8: Training instruction templates for RAG tuning with intermediate reasoning.

**Training RAG Answer templates.**   The RAG answer templates for different training datasets can be found in Table 9.

| Task | Answer Templates |
|------|-------------------|
| NQ | Question: {question} Answer: |
| Wizard of Wikipedia | Conversation: {question} Response: |
| FEVER | Return SUPPORTS if it is correct and return REFUTES if it is not correct. Fact: {question} Response: |
| MMLU | Question: {question} Options: {choices} Answer: |

Table 9: Training answer templates for RAG tuning with intermediate reasoning.

**Instructions to generate intermediate reasoning from Gemini-1.5-pro.**    The prompt that guides Gemini-1.5-pro for intermediate reasoning generation can be found in Table 10.

| Task | Prompts |
| --- | --- |
| NQ | Read the following documents relevant to the given question: {question} 
 {reference} 
 Please identify documents that are useful to answer the given question: {question}, 
 and explain how the contents lead to the answer: {answers}. 
 If none of the documents is aligned with the answer, 
 in that case, you have to explain the answer only based on your own knowledge, 
 without referring to the provided information. 
 Note that the question may be compositional and require intermediate analysis to deduce the final answer. 
 Make sure your response is grounded and provides clear reasoning details followed by a concise conclusion. |
| Wizard of Wikipedia | Read the following documents relevant to the given conversation: {question} 
 {reference} 
 Please identify documents that are useful to provide a response to a conversation: {question} 
 and explain how the contents lead to the response: {answers}. 
 If none of the documents is aligned with the response, 
 in that case, you have to explain the response only based on your own knowledge, 
 without referring to the provided information. 
 Make sure your response is grounded and provides clear reasoning details followed by a concise conclusion. |
| FEVER | Read the following documents relevant to the given question: {question} 
 {reference} 
 Please identify documents that are useful to verify a fact: {question} 
 (Return SUPPORTS if it is correct and return REFUTES if it is not correct.), 
 and explain how the contents lead to the answer: {answers}. 
 If none of the documents is aligned with the answer, 
 in that case, you have to explain the answer only based on your own knowledge 
 without referring to the provided information. 
 Make sure your response is grounded and provides clear reasoning details followed by a concise conclusion. |
| MMLU | Read the following documents relevant to the given question: {question} 
 {reference} 
 Please identify documents that are useful to answer the given question: {question} with options: {choices}, 
 and explain how the contents lead to the answer: {answers}. 
 If none of the documents is aligned with the answer, 
 in that case, you have to explain the answer only based on your own knowledge, 
 without referring to the provided information. 
 Make sure your response is grounded and provides clear reasoning details followed by a concise conclusion. |

Table 10: Prompts to guide Gemini-1.5-pro for intermediate reasoning generation.

## H.2 Evaluation settings

**Hyperparameters.** For all the compared LLMs, we conduct top-p sampling (p = 1) and the maximum number of generated token is set to be 256. For Gemma-2 series models, we use the huggingface inference pipeline. While for other series of LLMs, we utilize vLLM codebase for efficient generation.

**Evaluation RAG instruction templates.** The RAG instruction templates for different testing datasets can be found in Table 11.

| Task | Instruction Templates |
|------|----------------------|
| QA | Answer the question based on the given documents.
Please first provide an analysis with clear reasoning details of which documents are relevant to answer the question
Then output a concise answer to the question based on the analysis.
The following are given documents.
{reference} |
| Multi-hop | Answer the question based on the given documents.
Please first provide an analysis with clear reasoning details of which documents are relevant to answer the question
Then output a concise answer to the question based on the analysis.
The following are given documents.
{reference} |
| Long-form | Answer the question based on the given document.
Please first provide an analysis with clear reasoning details of which documents are relevant to answer the question
Then provide an in-depth long-form answer for the question (avoid only returning the answer) based on the analysis.
The following are given documents.
{reference} |
| Slot filling | Provided an answer to the given slot filling question based on the given document.
In the question, the words before and after [SEP] correspond to the head entity and relation respectively.
You are asked to output the tail entity corresponded to the given head entity and relation.
Please first provide an analysis with clear reasoning details of which documents are relevant to answer the question.
Then output a concise answer to the question based on the analysis.
The following are given documents.
{reference} |

Table 11: Testing instruction templates for RAG tuning with intermediate reasoning.

**Evaluation RAG answer templates.** The RAG answer templates for different testing datasets are all: "Question: {question}. Answer:"

## I DATA-AUGMENTED RAG CASE STUDIES

| Question | Which film features the Dawes Tomes Mousley Grubbs Fidelity Fiduciary Bank? |
|---|---|
| **Ground Truth** | Mary Poppins |
| **Retrieved Passages** | **Doc 1** (Title: "Fidelity Fiduciary Bank") Fidelity Fiduciary Bank ""Fidelity Fiduciary Bank"" is a song from Walt Disney's film ""Mary Poppins"", and it is composed by Richard M. Sherman and Robert B. Sherman. The song sung by the stodgy old bankers at the ""Dawes, Tomes, Mousely, Grubbs Fidelity Fiduciary Bank"", led by the ""Elder Mr. Dawes"" (Nackvid Keyed), to George Banks's two children, Jane and Michael, in an attempt to get Michael Banks to invest his tuppence in the bank. As the song continues the pressure is on George, a junior clerk at the bank, to sway Michael. When Michael finally, and ever so slightly, opens ...

**Doc 2** (Title: "Fidelity Fiduciary Bank") convince Michael to invest his money goes like this: has as its origins an essay by C. C. Turner titled 'Money London' in the book edited by G. R. Sims called ""Living London"" (London: 1903): Fidelity Fiduciary Bank ""Fidelity Fiduciary Bank"" is a song from Walt Disney's film ""Mary Poppins"", and it is composed by Richard M. Sherman and Robert B. Sherman. The song sung by the stodgy old bankers at the ""Dawes, Tomes, Mousely, Grubbs Fidelity Fiduciary Bank"", led by the ""Elder Mr. Dawes"" (Nackvid Keyed), to George Banks's two children, Jane and Michael, in an attempt to get ...
...
**Doc 40** (Title: "The Firm (soundtrack)") The Firm (soundtrack) The Firm is an album by American pianist Dave Grusin released in 1993, recorded for the GRP label. This album is a soundtrack to the Motion Picture ""The Firm"" directed by Sidney Pollack. It reached No. 131 on ""Billboard""'s Contemporary Jazz chart. This soundtrack was a 1993 Academy Award nominee for Best Original Score. All tracks composed by Dave Grusin; except where indicated In 2015 La-La Land Records issued a remastered and expanded edition, featuring the film score on disc one and the 1993 soundtrack album and bonus tracks on disc two. Italicised tracks on disc one ... |

Model Predictions

| Gemma-2-9B-Chat | Fidelity Fiduciary Bank |
|---|---|
| **Implicit RAG FT** | *Mary Poppins* (1964 film) |
| **RAG FT with Reasoning** | Documents 1 and 2 are useful to answer the question.

Both documents explicitly state that the song "Fidelity Fiduciary Bank" from the musical film *Mary Poppins* features the bank "Dawes, Tomes, Mousely, Grubbs Fidelity Fiduciary Bank".

Therefore, the film that features the mentioned bank is *Mary Poppins*. |

| Question | Which English chemist discovered the most elements? |
|---|---|
| **Ground Truth** | Humphry Davy |
| **Retrieved Passages** | **Doc 1** (Title: Chemistry) J.J. Thomson of Cambridge University discovered the electron and soon after the French scientist Becquerel as well as the couple Pierre and Marie Curie investigated the phenomenon of radioactivity. In a series of pioneering scattering experiments Ernest Rutherford at the University of Manchester discovered the internal structure of the atom and the existence of the proton, classified and explained the different types of radioactivity and successfully transmuted the first element by bombarding nitrogen with alpha particles. His work on atomic structure was improved on by his students, the Danish physicist Niels Bohr and Henry Moseley. The electronic theory of chemical ... 

 **Doc 2** (Title: Chemistry) weights. The development of the electrochemical theory of chemical combinations occurred in the early 19th century as the result of the work of two scientists in particular, J.J. Berzelius and Humphry Davy, made possible by the prior invention of the voltaic pile by Alessandro Volta. Davy discovered nine new elements including the alkali metals by extracting them from their oxides with electric current. British William Prout first proposed ordering all the elements by their atomic weight as all atoms had a weight that was an exact multiple of the atomic weight of hydrogen. J.A.R. Newlands devised an early table of ... 
 ... 
 **Doc 40** (Title: "History of chemistry") many had already applied the concept of atomism to chemistry. A major example was the ion theory of Svante Arrhenius which anticipated ideas about atomic substructure that did not fully develop until the 20th century. Michael Faraday was another early worker, whose major contribution to chemistry was electrochemistry, in which (among other things) a certain quantity of electricity during electrolysis or electrodeposition of metals was shown to be associated with certain quantities of chemical elements, and fixed quantities of the elements therefore with each other, in specific ratios. These findings, like those of Dalton's combining ratios, were early clues ... |

**Model Predictions**

| Gemma-2-9B-Chat | William Hyde Wollaston |
|---|---|
| **Implicit RAG FT** | William Ramsay |
| **RAG FT with Reasoning** | The best document to answer this question is **Doc 2(Chemistry)**. 
 This document states that **Humphry Davy** "discovered nine new elements including the alkali metals by extracting them from their oxides with electric current." 
 Therefore, the answer is *Humphry Davy*. 
 While other documents mention chemists and their discoveries, none of them contradict or provide a higher number of discovered elements than Davy's nine. This makes Doc 2 the most useful in identifying the English chemist who discovered the most elements. 
 It's important to note that this answer is based on Doc 2 alone. If other documents mentioned an English chemist who discovered more than nine elements, they would supersede the information in Doc 2. However, without that information, we can confidently say that based on these documents, Humphry Davy discovered the most elements among the English chemists mentioned. 
 **Conclusion:** Based on the given documents, *Humphry Davy* is the English chemist who discovered the most elements. |

# J    DATA-AUGMENTED RAG FINETUNING ON GEMMA-2-9B

In Figure 5, we illustrate the performance of implicit RAG finetuning on eight datasets with three different base models due to the space limitation. The whole results with Gemma-2-9B models can be found in Figure 11.

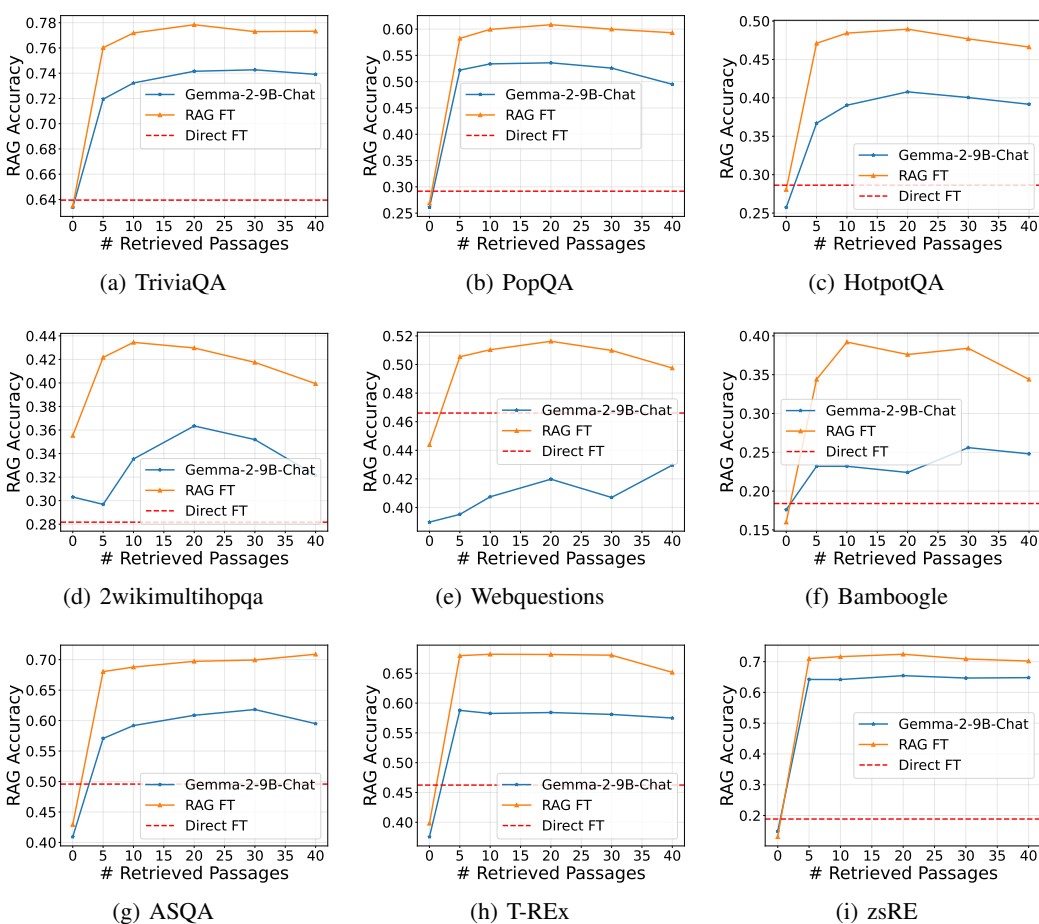

Figure 11: Generalization ability of LLMs fine-tuned with RAG-specific data (RAG FT). RAG FT consistently outperforms the chat LLM w. RAG and the model fine-tuned directly on question-answer pairs (Direct FT). This demonstrates the effectiveness of RAG FT in enabling the LLM to effectively extract knowledge from retrieved context on unseen tasks. Note that Direct FT is evaluated without retrieval to align with its training paradigm and all others are evaluated with retrieval augmentation. (LLM: Gemma-2-9B-Base)

In Figure 6, we show the power of RAG finetuning with intermediate reasoning on five datasets because of the space limitation. The whole results on all the nine datasets with Gemma-2-9B models can be found in Figure 12. Note that due to the computational complexity of inference with reasoning augmentation, results are shown for 1000 randomly-sampled queries for each dataset.

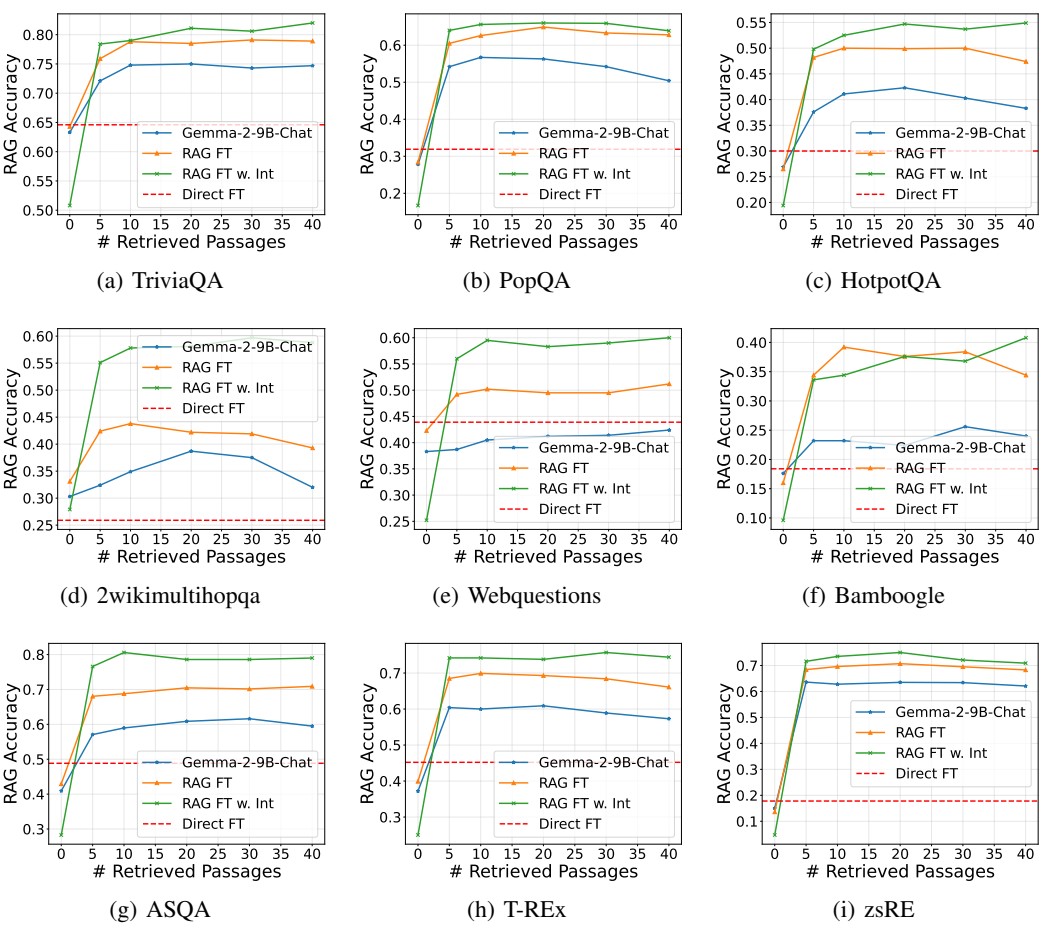

Figure 12: Evaluating the impact of intermediate reasoning on the performance of RAG-tuned LLMs. Results demonstrate that fine-tuning with an intermediate reasoning step (RAG FT w. Int) leads to further improvements compared to implicit RAG fine-tuning (RAG FT) and direct fine-tuning (Direct FT). Direct FT is evaluated without retrieval to align with its training paradigm and all others are evaluated with retrieval augmentation. Due to the computational complexity of inference with reasoning augmentation, results are shown for 1000 randomly-sampled queries from each dataset. (LLM: Gemma-2-9B-Base)

# K  DATA-AUGMENTED RAG FINETUNING ON MISTRAL-NEMO-12B

In addition to the comprehensive data-augmented RAG fine-tuning results with three different base LLMs reported in Section 5, we also would like to show the results specifically with the Mistral-Nemo-12B models in Figure 13.

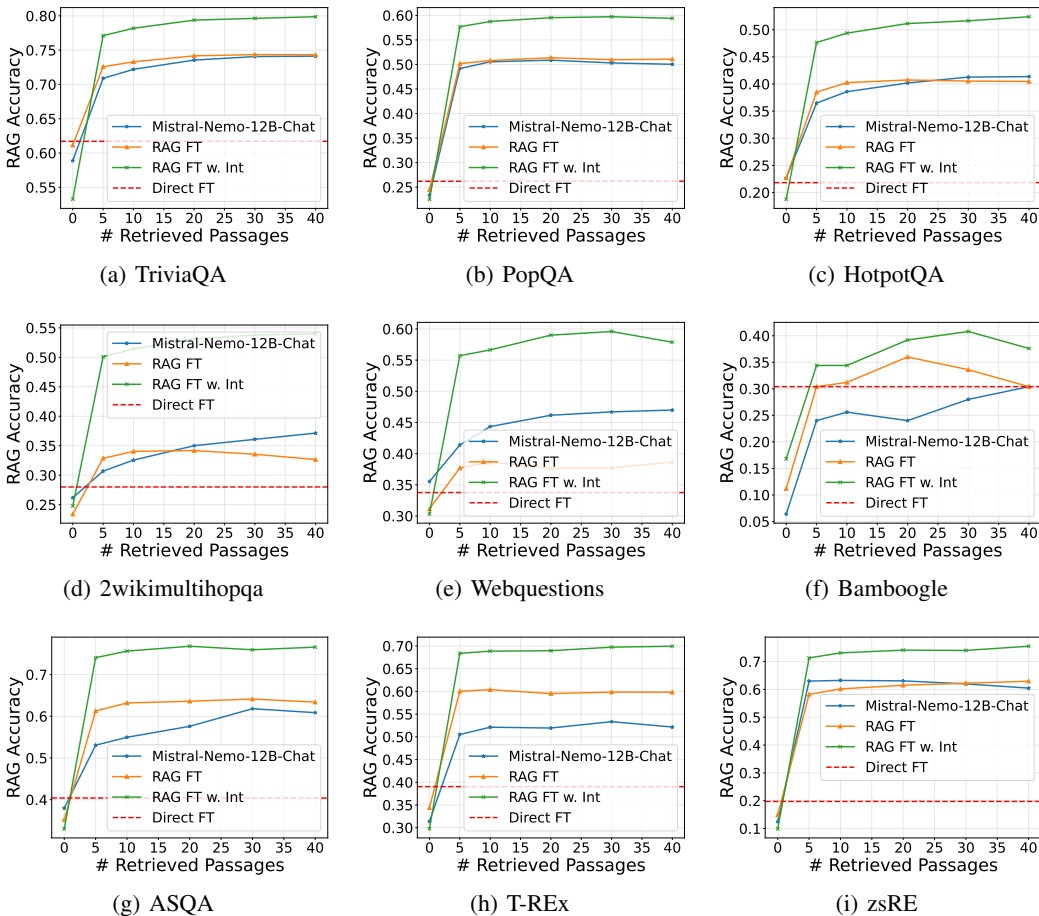

Figure 13: Evaluating RAG-specific tuning with Mistral-Nemo-12B models. Results demonstrate that fine-tuning with an intermediate reasoning step (RAG FT w. Int) leads to further improvements compared to implicit RAG fine-tuning (RAG FT), while implicit RAG fine-tuning outperforms LLMs without RAG-specific tuning (Mistral-Nemo-12B-Chat) and direct fine-tuning (Direct FT). Direct FT is evaluated without retrieval to align with its training paradigm and all others are evaluated with retrieval augmentation. (LLM: Mistral-Nemo-12B-Base)

## L    DATA-AUGMENTED RAG FINETUNING ON GEMINI-1.0-PRO

In addition to the comprehensive data-augmented RAG fine-tuning results with three different base LLMs reported in Section 5, we also would like to show the results specifically with the Gemini-1.0-Pro models in Figure 14. Due to the Gemini-1.0-Pro API call credit limitation, we random sample 1000 queries for each dataset.

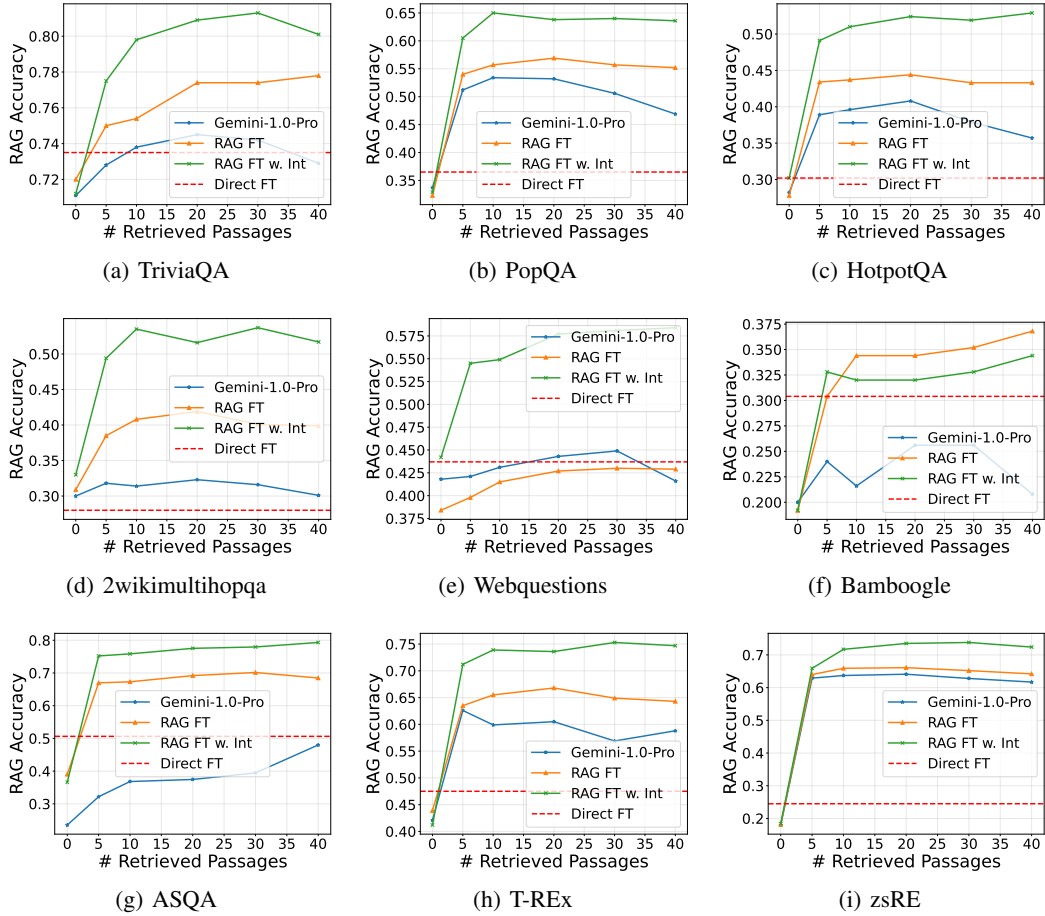

Figure 14: Evaluating RAG-specific tuning with Gemini-1.0-Pro models. Results demonstrate that fine-tuning with an intermediate reasoning step (RAG FT w. Int) leads to further improvements compared to implicit RAG fine-tuning (RAG FT), while implicit RAG fine-tuning outperforms LLMs without RAG-specific tuning (Gemini-1.0-Pro) and direct fine-tuning (Direct FT). Direct FT is evaluated without retrieval to align with its training paradigm and all others are evaluated with retrieval augmentation. Due to the Gemini-1.0-Pro API call credit limitation, results are shown for 1000 randomly-sampled queries from each dataset. (LLM: Gemini-1.0-Pro)

## M    TRAINING DATA SCALING AND RAG PERFORMANCE.

| Number of Retrieval Passages | 5k | 20k | 50k | 200k |
|---|---|---|---|---|
| 10 | 0.5942 | 0.5925 | 0.6058 | 0.6277 |
| 20 | 0.5909 | 0.5925 | 0.6078 | 0.6294 |
| 30 | 0.5787 | 0.5792 | 0.6072 | 0.6150 |
| 40 | 0.5582 | 0.5582 | 0.5859 | 0.5983 |
| Avg. | 0.5805 | 0.5806 | 0.6017 | 0.6176 |

Table 14: Impact of RAG-specific training data scale on LLM performance in RAG.

To investigate the influence of the size of the training data on the effectiveness of RAG-specific tuning, we fine-tune the Gemma-2-9B-Base model using varying amounts (5k to 200k samples) of mixed training data from NQ, WoW, Fever, and MMLU. Table 14 presents the evaluation results on the NQ dataset, demonstrating a clear positive correlation between the scale of training data and the performance of the resulting LLM in RAG. Increasing the amount of training data consistently leads to improved accuracy, highlighting the benefits of leveraging larger datasets for fine-tuning LLMs in RAG applications.

## N    RAG-SPECIFIC TUNING DATA INSIDE SFT MIXTURES

| Dataset | base | SFT only | SFT + RAG-FT |
|---|---|---|---|
| MT-Bench | 2.3125 | 5.8969 | 5.6031 |
| NQ | 0.2105 | 0.5687 | 0.6033 |
| TriviaQA | 0.4940 | 0.7155 | 0.7481 |

Table 15: Combining RAG-specific data with general SFT data for enhanced LLM performance in RAG.

Having established the effectiveness of RAG-specific fine-tuning for improving LLM performance in RAG tasks, we now investigate whether combining RAG-specific data with general SFT data can further enhance performance while preserving the LLM's general capabilities (*e.g.*, reasoning and long-form generation), as a way to assess the potential of the proposed tuning methods to be useful for construction of foundation models. We train the Gemma-2-9B model using two different strategies: (1) SFT data only: The LLM is trained solely on general SFT data (Ultrachat 200k). (2) SFT data + RAG-specific data: The LLM is trained on a combination of Ultrachat 200k and 50k RAG-specific data (the same data used in Figure 5). We evaluate the resulting models on MT-Bench to assess their general language capabilities and on NQ and TriviaQA to measure their RAG performance.

Table 15 presents the results, demonstrating that incorporating RAG-specific data into the SFT process can significantly improve the LLM's performance on RAG tasks while maintaining its performance on general language tasks. This finding suggests that combining task-specific and general-purpose data during fine-tuning can be a viable strategy for enhancing LLMs in specialized applications without compromising their overall capabilities.

## O  CASE STUDIES OF FAILURE QUERIES

In this section, we conduct case studies for queries which can be correctly answered by RAG with small number of retrieved passage (1), but are wrongly answered by RAG with larger number of retrieved passages (5).

**Case 1:**

| Question | When do primary ossification centers appear in an embryo? |
|---|---|
| **Ground Truth** | Prenatal development |
| **RAG with 1 retrieved passage** | |
| **Prediction** | Prenatal development |
| **Retrieved Passages** | **Doc 1** (Title: "Ossification center") of the same row also becomes calcified, and transverse bars of calcified substance stretch across from one calcareous column to another. Thus there are longitudinal groups of the cartilage cells enclosed in oblong cavities, the walls of which are formed of calcified matrix which cuts off all nutrition from the cells; the cells, in consequence, atrophy, leaving spaces called the primary areolæ. There are two types of ossification centers – primary and secondary. A primary ossification center is the first area of a bone to start ossifying. It usually appears during prenatal development in the central part of each developing ... |
| **RAG with 5 retrieved passages** | |
| **Prediction** | Fetal development |

| Retrieved Passages | **Doc 1** (Title: "Ossification center") of the same row also becomes calcified, and transverse bars of calcified substance stretch across from one calcareous column to another. Thus there are longitudinal groups of the cartilage cells enclosed in oblong cavities, the walls of which are formed of calcified matrix which cuts off all nutrition from the cells; the cells, in consequence, atrophy, leaving spaces called the primary areolæ. There are two types of ossification centers – primary and secondary. A primary ossification center is the first area of a bone to start ossifying. It usually appears during prenatal development in the central part of each developing ... |
|---|---|
| | **Doc 2** (Title: "Ossification center") bone. In long bones the primary centers occur in the diaphysis/shaft and in irregular bones the primary centers occur usually in the body of the bone. Most bones have only one primary center (e.g. all long bones) but some irregular bones such as the os coxa (hip) and vertebrae have multiple primary centers. A secondary ossification center is the area of ossification that appears after the primary ossification center has already appeared – most of which appear during the postnatal and adolescent years. Most bones have more than one secondary ossification center. In long bones, the secondary centres appear in ... |
| | **Doc 3** (Title: "Primary bone") Primary bone Primary bone is the first bone tissue that appears in embryonic development and in fracture repair. It is characterized by its random position of collagen fibers. In most places in adults this tissue is replaced by secondary bone tissue except, for example, near the sutures of calvara or tooth sockets. The secondary bones have lower amounts of osteocytes so primary bone is much more easily penetrated by x-ray. Primary bone or the primary ossification center is the beginning of the bone building process during the first trimester. Calcified cartilage is basophilic and new bone being made is more ... |
| | **Doc 4** (Title: Ossification) small opening in the diaphysis. It invades the primary center of ossification, bringing osteogenic cells (osteoblasts on the outside, osteoclasts on the inside.) The canal of the nutrient foramen is directed away from more active end of bone when one end grows more than the other. When bone grows at same rate at both ends, the nutrient artery is perpendicular to the bone. Most other bones (e.g. vertebrae) also have primary ossification centers, and bone is laid down in a similar manner. Secondary centers The secondary centers generally appear at the epiphysis. Secondary ossification mostly occurs after birth (except for ... |
| | **Doc 5** (Title: Bone) development of the periosteum. Endochondral ossification occurs in long bones and most other bones in the body; it involves the development of bone from cartilage. This process includes the development of a cartilage model, its growth and development, development of the primary and secondary ossification centers, and the formation of articular cartilage and the epiphyseal plates. Endochondral ossification begins with points in the cartilage called ""primary ossification centers."" They mostly appear during fetal development, though a few short bones begin their primary ossification after birth. They are responsible for the formation of the diaphyses of long bones, short bones and ... |

**Case 2:**

| Question | What is the final season of downton abbey? |
|---|---|
| Ground Truth | The sixth series |
| **RAG with 1 retrieved passage** | |
| Prediction | The sixth series |

| Retrieved Passages | **Doc 1** (Title: "Downton Abbey") both ITV and PBS, and subsequently became the most successful British costume drama series since the 1981 television serial of ""Brideshead Revisited"". On 26 March 2015, Carnival Films and ITV announced that the sixth series would be the last. It aired on ITV between 20 September 2015 and 8 November 2015. The final episode, serving as the annual Christmas special, was broadcast on 25 December 2015. A film adaptation was confirmed on 13 July 2018. The first series, comprising seven episodes, explores the lives of the fictional Crawley family, the hereditary Earls of Grantham, and their domestic servants. The storyline ... |
| --- | --- |

| **RAG with 5 retrieved passages** ||
| --- | --- |
| **Prediction** | Between 20 September 2015 and 8 November 2015 |
| Retrieved Passages | **Doc 1** (Title: "Downton Abbey") both ITV and PBS, and subsequently became the most successful British costume drama series since the 1981 television serial of ""Brideshead Revisited"". On 26 March 2015, Carnival Films and ITV announced that the sixth series would be the last. It aired on ITV between 20 September 2015 and 8 November 2015. The final episode, serving as the annual Christmas special, was broadcast on 25 December 2015. A film adaptation was confirmed on 13 July 2018. The first series, comprising seven episodes, explores the lives of the fictional Crawley family, the hereditary Earls of Grantham, and their domestic servants. The storyline ...

**Doc 2** (Title: "Downton Abbey (film)") Downton Abbey (film) Downton Abbey is an upcoming British historical period drama film, written by Julian Fellowes and directed by Michael Engler. It is a continuation of the television series of the same name, created by Fellowes, that ran on ITV from 2010 to 2015. The film is scheduled to be released on 13 September 2019 in the United Kingdom, and one week later in the United States on 20 September. The film is a follow-up to the television series of the same name, which ended its original run in December 2015 after 52 episodes. In April 2016, it was ...

**Doc 3** (Title: "Downton Abbey") Focus Features and Universal Pictures International. The film is scheduled for a UK release on 13 September 2019, with the US following one week later on 20 September 2019. Downton Abbey Downton Abbey is a British historical period drama television series set in the early 20th century, created by Julian Fellowes. The series first aired on ITV in the United Kingdom on 26 September 2010, and in the United States on PBS, which supported production of the series as part of its ""Masterpiece Classic"" anthology, on 9 January 2011. The series, set in the fictional Yorkshire country estate of Downton ...

**Doc 4** (Title: "Downton Abbey") Downton Abbey Downton Abbey is a British historical period drama television series set in the early 20th century, created by Julian Fellowes. The series first aired on ITV in the United Kingdom on 26 September 2010, and in the United States on PBS, which supported production of the series as part of its ""Masterpiece Classic"" anthology, on 9 January 2011. The series, set in the fictional Yorkshire country estate of Downton Abbey between 1912 and 1926, depicts the lives of the aristocratic Crawley family and their domestic servants in the post-Edwardian era—with the great events in history having an effect ...

**Doc 5** (Title: "Downton Abbey") Mary, but she puts him off when Lady Grantham becomes pregnant. Cora miscarries after O'Brien, believing she was to be fired, retaliates by kicking a broken bar of soap near the bathtub steps, making Cora slip and fall, killing her unborn child which had been a boy. The series ends just after the assassination of Archduke Franz Ferdinand and the outbreak of the First World War in July 1914. The second series comprises eight episodes and runs from the Battle of the Somme in 1916 to the 1918 Spanish flu pandemic. During the war, Downton Abbey is temporarily converted into ... |

**Case 3:**

| Question | When does the movie jeepers creepers come out? |
| --- | --- |

| Ground Truth | September 26, 2017 |
|---|---|

**RAG with 1 retrieved passage**

| Prediction | September 26, 2017 |
|---|---|
| **Retrieved Passages** | **Doc 1** (Title: "Jeepers Creepers 3") Jeepers Creepers 3 Jeepers Creepers 3 is a 2017 American horror film written and directed by Victor Salva and the third ""Jeepers Creepers"" film, taking place in between ""Jeepers Creepers"" and ""Jeepers Creepers 2."" Jonathan Breck reprises his role as the Creeper. Gina Philips returns in a cameo as Trish Jenner, her first return to the series since the original film. The film was shown in theaters on September 26, 2017 in what was originally announced as a one-night-only showing and was then shown again on October 4, 2017. A shuriken flies through the air and hits a wooden post ... |

**RAG with 5 retrieved passages**

| Prediction | Between 20 September 2015 and 8 November 2015 |
|---|---|
| **Retrieved Passages** | **Doc 1** (Title: "Jeepers Creepers 3") Jeepers Creepers 3 Jeepers Creepers 3 is a 2017 American horror film written and directed by Victor Salva and the third ""Jeepers Creepers"" film, taking place in between ""Jeepers Creepers"" and ""Jeepers Creepers 2."" Jonathan Breck reprises his role as the Creeper. Gina Philips returns in a cameo as Trish Jenner, her first return to the series since the original film. The film was shown in theaters on September 26, 2017 in what was originally announced as a one-night-only showing and was then shown again on October 4, 2017. A shuriken flies through the air and hits a wooden ...

**Doc 2** (Title: "Jeepers Creepers 2") spot still held by ""Jeepers Creepers"". Allowing for films that had been released prior to Labor Day, ""Jeepers Creepers 2"" holds the 9 spot after the 2015 Labor Day four-day weekend. In September 2015, ""Jeepers Creepers 3"" was officially greenlit. The film was slated to begin filming in April 2016 until production was halted when Victor Salva was boycotted from filming in Canada for his criminal past The film was eventually released in a one-night-only showing on September 26, 2017, 14 years after the release of ""Jeepers Creepers 2"". It grossed $2.3 million in theaters. Jeepers Creepers 2 Jeepers Creepers ...

**Doc 3** (Title: "Jeepers Creepers 3") been planned. This more closely replicates the Florida setting of the original film. Principal photography began on February 15, 2017 in Baton Rouge, Louisiana. On April 4, 2017, one of the cameramen of the production revealed on their social networks that filming had by then been completed. During an interview for the Edmond Sun, Justin Hall specifically revealed that the film would be released on September 4, 2017, but nothing was confirmed by Salva or the studio. On August 16, 2017, the AMC Theatres website stated that ""Jeepers Creepers 3"" would open on September 26, 2017. On August 29, 2017, ...

**Doc 4** (Title: "Jeepers Creepers (2001 film)") McCoy"" AKA ""Roach"" who is a car thief and regular in the Poho County jail. In the second film, he portrays ""Coach Dwayne Barnes"". On September 11, 2015, ""Jeepers Creepers 3"" was officially greenlit, with a planned 2017 release. Victor Salva returns as director, Jonathan Breck returns as The Creeper, and Gina Philips returns as Trish Jenner, her first screen role in five years. Production was halted in 2016 until it resumed in February 2017, and completed in April. The film opened for what was said would be only a one-night showing on September 26, 2017; it was then shown ...

**Doc 5** (Title: "Jeepers Creepers (2001 film)") Jeepers Creepers (2001 film) Jeepers Creepers is a 2001 American-German horror film written and directed by Victor Salva. The film takes its name from the 1938 song ""Jeepers Creepers"", which is featured in the film. Francis Ford Coppola executive produced, and the film stars Gina Philips, Justin Long, Jonathan Breck, and Eileen Brennan. Philips and Long play two older siblings who become the targets of a demonic creature (Breck) in rural Florida. Trish Jenner (Philips) and her brother Darry (Long) are traveling home from college for spring break. As they drive through the Florida countryside, an old rusty truck ... |

Our analysis reveals no clear pattern distinguishing queries that can be effectively answered with a small number of retrieved passages but fail when the retrieval set size increases. We hypothesize that these failure cases arise from complex interactions between the query, retriever, and knowledge corpus. Specifically, for a given query, if the retriever successfully identifies relevant information with a small retrieval top-$k$, but increasing $k$ primarily introduces noise, the likelihood of failure in answering the query correctly with a larger retrieval set increases significantly.

## P    COST-BENEFIT ANALYSIS OF BASIC RERANKING AND OUR PROPOSED METHODS

To provide a comprehensive assessment, we analyze the cost-benefit trade-offs between basic reranking, retrieval reordering (Ours), and RAG-specific LLM tuning (Ours).

**Basic reranking.**    **Cost**: Reranking involves processing every query in each testing dataset with a computationally expensive cross-encoder model. This process can be time-consuming, especially for large datasets. In our experiments, reranking the top 200 passages on the TriviaQA dataset (over 11k samples) took over 1.5 hours (with bge-reranker-large (Li et al., 2023)) using DataParallel and 8 H100s (while retrieval only (without reranking) with e5 and gpu-based approximated nearest neighbor search - faiss-gpu - only takes 10 mins). This cost scales linearly with dataset size, making it a significant bottleneck for large-scale applications. **Benefit**: Reranking can improve retrieval accuracy, leading to better overall RAG performance, as demonstrated in our results above.

**Retrieval reordering (Ours).**    **Cost**: This method is designed to be plug-and-play and can be seamlessly integrated with any existing RAG pipeline (retriever, reranker, and generator) without incurring additional computational costs; **Benefit**:It consistently improves RAG performance, particularly when using a larger number of retrieved passages. Importantly, it is both retriever-agnostic and reranker-agnostic, demonstrating its versatility and broad applicability (as shown in Figure 4).

**RAG-specific LLM tuning (Ours).**    **Cost**: Training an LLM requires computational resources and time. In our experiments, achieving a well-trained checkpoint took approximately 2 hours using DeepSpeed zero3 and 8 H100s. **Benefit**: This method yields substantial and consistent improvements in the LLM's ability to handle RAG scenarios across different numbers of retrieved passages (as shown in the table above). Crucially, this training is a one-time cost; the resulting LLM can be deployed for any testing dataset without incurring further computational overhead during inference (retrieval with e5 and gpu-based approximated nearest neighbor search - faiss-gpu - only takes 10 mins on TriviaQA, while reranking with bge-reranker-large (Li et al., 2023) takes more than 1.5 hours).

It's important to highlight that these three approaches are orthogonal and address different aspects of the RAG system: **Basic reranking** focuses on improving the quality of retrieved passages. **Retrieval reordering** optimizes the interaction between the retriever and the LLM generator. **RAG-specific LLM tuning** enhances the generator's capabilities for handling retrieved information. This inherent complementarity suggests that combining these techniques can lead to further performance gains in RAG systems.

## Q    INTUITION BEHIND RETRIEVAL REORDERING

The intuition behind retrieval reordering is that, on average, hard negatives are ranked lower than relevant passages. To illustrate this, we conducted experiments on the NQ dataset with both e5 and BM25 retrievers, analyzing the top-$k$ retrieved passages ($k$ = 5/10/20/50/100). We calculated the average ranking of both relevant passages and hard negatives, first averaging at the query level and then across all queries. The results are shown in Table 19 and Table 20.

Our findings consistently show that the average ranking of relevant passages is higher than that of hard negatives for both retrievers and across all top-k values. This observation supports the rationale behind our retrieval reordering strategy: by prioritizing higher-ranked passages, we increase the likelihood that the LLM focuses on relevant information.

| # Retrieved Passages (Top-$k$) | 5 | 10 | 20 | 50 | 100 |
|---|---|---|---|---|---|
| Avg. Relevant Passage Ranking | 2.497 | 4.171 | 7.406 | 16.638 | 30.826 |
| Avg. Hard Negative Ranking | 3.426 | 6.157 | 11.429 | 27.044 | 52.801 |

Table 19: Average rankings of relevant passages and hard negatives on NQ for different numbers of retrieved passages (retriever: e5).

| # Retrieved Passages (Top-$k$) | 5 | 10 | 20 | 50 | 100 |
|---|---|---|---|---|---|
| Avg. Relevant Passage Ranking | 2.696 | 4.808 | 8.609 | 19.615 | 36.252 |
| Avg. Hard Negative Ranking | 3.189 | 5.784 | 10.931 | 26.262 | 51.719 |

Table 20: Average rankings of relevant passages and hard negatives on NQ for different numbers of retrieved passages (retriever: BM25).

Furthermore, it's important to emphasize that the benefits of reordering extend beyond mitigating the impact of hard negatives. By placing higher-relevance passages in more prominent positions, we simultaneously reduce the influence of lower-relevance passages, leading to a more effective utilization of the retrieved information by the LLM.

# R    EXPERIMENTS ON MEDICINE DOMAIN RAG

We have expanded our analysis to include RAG experiments on PubMedQA and BioASQ (Xiong et al., 2024), utilizing PubMed [§] as the retrieval corpus.

| # Retrieved Passages (Top-$k$) | 0 | 1 | 5 | 10 | 20 |
|---|---|---|---|---|---|
| PubMedQA | 0.314 | **0.592** | 0.41 | 0.43 | 0.46 |
| BioASQ | 0.7524 | 0.7638 | **0.7767** | **0.7735** | 0.7330 |

| # Retrieved Passages (Top-$k$) | 30 | 40 | 50 | 100 | 150 |
|---|---|---|---|---|---|
| PubMedQA | 0.416 | 0.38 | 0.394 | 0.32 | 0.354 |
| BioASQ | 0.7120 | 0.6974 | 0.6505 | 0.6845 | 0.6942 |

Table 21: Performance metrics for PubMedQA and BioASQ datasets across different numbers of retrieved passages. (LLM: Mistral-Nemo-12B-Instruct)

These new experiments confirm the presence of the "inverted U-curve phenomenon" with mistral-nemo-instruction as the generator across both datasets (with PubMed as the knowledge base and e5 as the retriever), further reinforcing our observations.

# S    LIMITATIONS AND FUTURE WORK

In this paper, we first identify the inverse-U performance curve exhibited by Retrieval-Augmented Generation (RAG) systems as the number of retrieved passages increases. We then conduct a systematic analysis to understand the underlying causes of this phenomenon and propose three approaches to enhance the performance of long-context large language models (LLMs) in RAG applications. However, due to page limitations, we acknowledge that certain problems and solutions are only briefly addressed and warrant deeper exploration.

First, the robustness of Gemini-1.5-Pro to random negatives and its lack of an observable inverse-U curve remain insufficiently explained. We hypothesize that the quality and characteristics of its pretraining and post-training datasets play a critical role. Unfortunately, the absence of publicly available information regarding the training data for these LLMs precludes definitive conclusions at this time. We recognize this limitation and encourage further research into the impact of training data on retrieval robustness as an important future direction.

---

[§]https://pubmed.ncbi.nlm.nih.gov/

Second, we propose a straightforward yet effective solution: retrieval reordering. By strategically positioning high-scoring passages at the beginning and end of the input sequence, we leverage the known positional biases of LLMs, which tend to prioritize information in these regions. This introduces the concept of "position engineering" in RAG systems, highlighting the critical role of passage ordering. While this work focuses on demonstrating the efficacy of this approach, we acknowledge the potential for exploring alternative reordering strategies and suggest this as a promising avenue for future research.

Finally, we present two tuning-based methods to enhance the robustness of LLMs in RAG applications: (1) implicit RAG-specific tuning, and (2) RAG-specific tuning with intermediate reasoning steps. These methods aim to improve the alignment of LLMs with RAG tasks and their ability to handle noisy or redundant retrievals. Future work could explore more fine-grained and multi-step reasoning chains to further refine RAG performance and robustness.

