# OpenReview forum: "Long-Context LLMs Meet RAG: Overcoming Challenges for Long Inputs in RAG"
_ICLR.cc/2025/Conference — ICLR 2025 Poster_

### Official Review · Reviewer_ua3n · 2024-10-29

**Soundness:** 3
**Presentation:** 3
**Contribution:** 3
**Rating:** 8
**Confidence:** 4

**Summary:**

This paper investigates the effect of increasing the length of the retrieved document list on the performance of RAG. They find that simply increasing the number of retrieved documents does not consistently improve the performance of RAG, and both training-free and training-based methods are proposed to solve this problem.

**Strengths:**

1. The writing is clear and easy to follow

2. Experimental results show the effectiveness of the proposed methods.

3. This paper makes significant contribution in the context of the rapid development of long-context LLM

**Weaknesses:**

In this rebuttal period, the authors address most of my concerns below

1. **Wrong research basis**: In Abstract and Introduction Sections, this paper's description of the current research status of the RAG community is wrong, which directly leads to the fact that the research basis of this paper is wrong, which is unacceptable. Specifically, in Lines 14-16 of Abstract "It is plausible to assume that a larger retrieval set would contain more relevant information (higher recall), that might result in improved performance" and Lines 46-47 of Introduction "Intuitively, longer context would allow for the inclusion of more retrieved passages, leading to higher recall and potentially improved performance". Both claims misrepresent the current state of research in the RAG community. Many studies in RAG have shown that increasing the number of retrieved documents cannot consistently improve the RAG performance due to the noise or incorrectness of the retrieved texts [1, 2 ,3, 4, 5]. This paper wrongly uses a problem that has been clarified in the RAG community as the motivation, which is unacceptable.

2. **Over-claiming of contributions**: As weakness 1 indicates that the robustness of RAG has been fully-studied in RAG [1,2,3,4,5] and shows irrelevant retrieved documents may interference the performance of RAG, the main contribution claimed in this paper that "we systematically analyze the use of long-context LLMs in RAG systems, specifically examining the impact of retrieved "hard negatives" on performance" cannot be seen as the contribution of this paper.

3. **Limited technical contribution**：This paper claims that they propose two methods to enhance the robustness of RAG on "hard negatives". One method is a training-free method based on retrieval reordering, this method is incremental based on Lost-in-the-Middle [6]. The other method is explicit tuning with intermediate reasoning for relevance identification, this is similar to RetRobust [1] without distinguishable points.

[1] Making Retrieval-Augmented Language Models Robust to Irrelevant Context

[2] Chain-of-note: Enhancing robustness in retrieval-augmented language models

[3] Corrective retrieval augmented generation

[4] Evaluation of Retrieval-Augmented Generation: A Survey

[5] Benchmarking large language models in retrieval-augmented generation

[6] Lost in the Middle: How Language Models Use Long Contexts

**Questions:**

Please see Weaknesses above.

---

> ### Author Response · Authors · 2024-11-20
> **Author Response (1/3)**
>
> We sincerely appreciate your insightful feedback and have carefully addressed your comments below.
>
> - **Wrong research basis.**
>
> We appreciate you pointing out those relevant references. We've carefully reviewed your feedback and revised our manuscript to ensure a comprehensive literature review and accurately position our contributions. We have modified the motivation and illustration in Abstract/Section 1 and included a comparison with the suggested previous works in Section 2.
>
> To be specific, (1) Lines 14-16 and 46-47: We've rephrased these lines to accurately reflect their basis in long-context LLM research [7][8], rather than implying broader RAG claims. (2) Section 1: We've clarified that prior work hasn't focused on evaluating long-context LLMs in real-world RAG scenarios, highlighting the novelty of our approach. (3) Contribution: We've sharpened the description of our analysis contribution in Section 1 to ensure accuracy.
>
> Importantly, we would also like to argue that none of the mentioned works have studied long-context LLMs with RAG and confirmed that “**under the newly emerged strong long-context LLM development [9][10][11]**, increasing the number of retrieved passages still cannot consistently improve the RAG performance due to the noise or incorrectness of the retrieved texts”. We note that the lines of research on long-context RAGs vs. noise-robust RAGs are relevant but not exactly aligned - the caveats and optimal bottlenecks might have notable differences. Increasing the number of retrieved passages not only increases the amount of noise in the context (lower precision and hard negatives) but also introduces more potentially relevant passages (higher recall as shown in Figure 2). Since more relevant passages could potentially benefit RAG performance, it is underexplored how the increased noise could still affect the RAG system. In addition, it is also not studied what is the correlation between the hard negatives and retriever capability (discussed in Section 3.3).  We clarify the contribution of analysis in Section 1 to reflect this.
>
> We provide a detailed comparisons with existing works mentioned by the reviewer to highlight the key distinctions:
>
> **Comparison with [1]**: While [1] focuses on **determining whether to use retrieved information at all** when the **entire** retrieved context is deemed irrelevant, our work explores a different dimension: **passage**-level relevance within **large-scale** retrieval scenarios.  Increasing the number of retrieved passages introduces a complex interplay of relevant and irrelevant passages, creating a unique challenge for LLMs in effectively utilizing this mixed information. Our research delves into this under-explored area, providing a comprehensive analysis and proposing three novel solutions to improve LLM performance in such scenarios.
>
> **Comparison with [2]**: [2] focuses on improving the robustness of Retrieval-Augmented Language Models (RALMs) in scenarios with a **small** number of retrieved passages (around 5) and relies on **synthetic** experiments for evaluation. In contrast, our work conducts a systematic empirical study of long-context RAG in **real-world** settings, specifically addressing the challenges posed by a **large** number of retrieved passages.  This distinction is crucial because processing a larger volume of retrieved information demands more advanced long-context capabilities from the LLM. While previous research has explored long-context LLMs in synthetic benchmarks, their performance in real-world, large-scale RAG scenarios remains under-explored. Our work fills this gap by providing a comprehensive analysis and offering three potential solutions to enhance long-context LLMs in such settings.
>
> **Comparison with [3]**: [3] proposes a filtering-based approach to identify and remove irrelevant passages from the retrieved context. Our work, however, focuses on improving the long-context LLM generators themselves within large-scale retrieval RAG systems. These two approaches are complementary and can be combined to further enhance the robustness of the overall RAG system.
>
> **Comparison with [4][5]**: [4] and [5] focus on evaluating RAG systems, with one providing a survey of evaluation techniques and the other introducing a new benchmark corpus. Our work differs by specifically investigating the behavior and improvement of long-context LLMs within RAG systems utilizing large-scale retrieved contexts.
>
> In summary, our work presents a novel and systematic empirical study of long-context LLMs in large-scale RAG scenarios. We offer unique insights into the challenges posed by increasing retrieval scale and propose effective solutions to enhance LLM performance in these settings. This focus differentiates our contributions from existing research, which primarily concentrates on different aspects of RAG, such as handling small retrieved contexts, filtering irrelevant information, or evaluating overall system performance.

---

> ### Author Response · Authors · 2024-11-20
> **Author Response (2/3)**
>
> - **Over-claiming of contributions.**
>
> Thank you for the comment. While we have already detailed the specific distinctions between our work and the five suggested papers, we want to emphasize the unique contributions of **our analysis** compared to the existing literature:
>
> **(1) Discovery of the inverted U-curve (Section 3.1)**: We are the first to systematically investigate the performance of various LLMs in RAG systems as the number of retrieved passages increases, leading to the discovery of the "inverted U-curve" phenomenon. While prior work like "lost-in-the-middle" [6] has explored the limitations of long-context LLMs, our research connects this to retrieval precision/recall in real-world RAG scenarios, providing a novel perspective on the interplay between retrieval and generation.
>
> **(2) Analysis of the inverted U-curve phenomenon (Section 3.2)**: We acknowledge the existing research on irrelevant retrieved information in RAG. However, we argue that simply focusing on noise is insufficient to explain the inverted U-curve. The initial performance increase with more retrieved passages indicates a complex interplay between relevant and irrelevant information. As the number of retrieved passages grows, the LLM must contend with both the benefits of increased relevant information and the detrimental effects of noise. This dynamic is not fully captured by existing work on noise in RAG, highlighting the distinct contribution of our research on long-context RAG. Moreover, we observe variations in the inverted U-curve across different retrievers, demonstrating that precision alone is an inadequate metric for quantifying the impact of irrelevant retrieved information.
>
> **(3) Deep understanding of the hard negatives (Section 3.3)**: Through synthetic experiments, we uncover a novel finding: the strength of the retriever directly correlates with the difficulty LLMs face in handling the corresponding hard negatives. This important insight, not previously identified in the literature, sheds new light on the complex relationship between retriever strength and LLM robustness in RAG systems.

---

> ### Author Response · Authors · 2024-11-20
> **Author Response (3/3)**
>
> - **Limited technical contribution.**
>
> Thank you for your comment. We would like to highlight the key technical contributions of our proposed methods:
>
> (1) **Retrieval reordering**: While this method is indeed simple and effective, its advantages extend beyond its ease of implementation. It offers high efficiency, significant performance gains, and seamless integration into existing RAG pipelines due to its plug-and-play nature.  Importantly, the impact of retrieval order has not been extensively studied in the context of RAG. By introducing this method, we aim to draw attention to the importance of **position engineering** and encourage further research in this area.
>
> (2) **Tuning-based methods**: We introduce two novel tuning-based methods to enhance the performance of long-context LLMs in RAG: an implicit tuning method and a method incorporating intermediate reasoning. These methods offer several advantages over existing approaches like RetRobust:  **Granularity of Focus**: RetRobust primarily addresses scenarios where the entire retrieved context is irrelevant, focusing on deciding whether to utilize retrieved information at all. It predominantly considers retrieving a *single passage* and operates at the *retrieval-level*. In contrast, our work explores *large-scale retrieval* and delves into *passage-level* relevance, enabling finer-grained control over information utilization.  **Generalization**: RetRobust primarily focuses on in-domain tuning, while our methods demonstrate strong out-of-domain generalization capabilities. This highlights the ability of our RAG-specific tuning to improve the inherent RAG capabilities of LLMs, leading to enhanced performance on unseen data. **Intermediate Reasoning**:  Our approach explicitly trains LLMs to perform intermediate reasoning before generating the final answer. This differs significantly from RetRobust, which focuses on directly tuning the LLM to answer the question without an explicit reasoning step.
>
> To further demonstrate the effectiveness of our methods, we conducted additional experiments comparing them with RetRobust [1]. The results show that our proposed techniques outperform RetRobust by a significant margin.
>
> NQ
>
> | # retrieved psg | 1 | 5 | 10 | 20 |
> |--|--|---|--|--|
> | RetRobust (NLI) | 0.3864 | 0.3485 | 0.4380 | 0.4573 |
> | RetRobust (tuning) | 0.4781 | 0.5042 | 0.5017 | 0.4814 |
> | RAG implicit tuning (Ours) | 0.4983 | 0.5640 | 0.5903 | 0.6003 |
> | RAG tuning with reasoning (Ours) | 0.5673 | 0.6659 | 0.6792 | 0.6928 |
>
> PopQA
>
> | # retrieved psg | 1 | 5 | 10 | 20 |
> |--|--|--|--|--|
> | RetRobust (NLI) | 0.2641 | 0.2603 | 0.4063 | 0.4197 |
> | RetRobust (tuning) |  0.4727 | 0.5166 | 0.5027 | 0.4544 |
> | RAG implicit tuning (Ours) | 0.4820 | 0.5524 | 0.5599 | 0.5606 |
> | RAG tuning with reasoning (Ours) | 0.5234 | 0.6102 | 0.6207 | 0.6292 |
>
>
> (3) **Comprehensive study of RAG-specific LLM tuning**: LLM tuning for RAG purposes has a lot of design choices including LLM, context length limit, retriever, training/evaluation dataset etc.. We conduct a systematic study of different design choice components in Section 6 and Appendix M and N to provide design hints for researchers who are interested in implementing RAG-specific LLM tuning for their work or research.
>
>
> [1] Making Retrieval-Augmented Language Models Robust to Irrelevant Context
>
> [2] Chain-of-note: Enhancing robustness in retrieval-augmented language models
>
> [3] Corrective retrieval augmented generation
>
> [4] Evaluation of Retrieval-Augmented Generation: A Survey
>
> [5] Benchmarking large language models in retrieval-augmented generation
>
> [6] Lost in the Middle: How Language Models Use Long Contexts
>
> [7] Retrieval Augmented Generation or Long-Context LLMs? A Comprehensive Study and Hybrid Approach
>
> [8] Can Long-Context Language Models Subsume Retrieval, RAG, SQL, and More?
>
>
> [9] Gemini 1.5: Unlocking multimodal understanding across millions of tokens of context
>
> [10] The Llama 3 Herd of Models
>
> [11] Mistral NeMo

---

> ### Comment · Reviewer_ua3n · 2024-11-22
> **Reviewer‘s response to rebuttal**
>
> Thanks for your rebuttal, I have read your response thoroughly, but I do not think it addresses my concerns. Specifically:
>
> 1. In your response about wrong research basis, you argue that an important difference between your RAG for long-context and previous studies on the robustness of RAG to noise is that longer contexts not only introduce more noise, but may also introduce more positive passages. But this point is still essentially consistent with the previous study about the robustness of RAG to irrelevant documents. **The complex relationship between relevant and irrelevant passages brought about by long context** is essentially still a robustness issue of LLM in RAG. Imagine that an LLM model is faced with a retrieved list containing multiple passages, which contains both relevant and irrelevant passages. As long as the robustness of the LLM is high enough, it can ensure that the interference of irrelevant passages can be avoided. In this case, the increase in the length of the retrieved list will bring a stable performance improvement. In this case, the question you are studying is no longer valid. **Therefore, the seemingly complex relationship you emphasize is essentially still the problem of LLM's robustness to irrelevant passages in RAG. You have essentially taken a problem that has already been well studied in previous work, wrapped it in a seemingly more complex guise to highlight the contribution of your work, which is not encouraged.**
>
> 2. In your response about contribution, this still ties back to the point I emphasized earlier: this is fundamentally an issue of RAG's robustness to retrieved documents. The inverted U-shaped curve you highlighted is merely a repackaging of this issue. Specifically, in the curve you described, as the length of the retrieved list increases, RAG's performance first improves and then declines. This is essentially because the positive samples are ranked higher in the list, while the later portions predominantly consist of negative samples. Initially, as the retrieved list grows, more positive samples are included, leading to an improvement in RAG's performance. However, as the list continues to grow, more negative passages are introduced in the later ranks, which interfere with the LLM and cause a decline in performance. Isn't this fundamentally a matter of RAG's robustness?
>
> 3. Regarding your point on the "Deep understanding of the hard negatives," specifically that "the strength of the retriever directly correlates with the difficulty LLMs face in handling the corresponding hard negatives," do you really think this is a point worth emphasizing? Isn’t it rather self-evident? For a negative sample that can confuse a stronger retriever (and thus qualify as a harder negative for that retriever), it naturally suggests a higher likelihood of confusing the LLM as well, leading to reduced robustness of RAG against it. This seems like an obvious issue—why would you frame it as a "novel" contribution?
>
> So I still believe that this paper does not meet the standards of ICLR, and I will not revise my review.

---

> > ### Author Response · Authors · 2024-11-22
> > **Round2 Author Response (2/2)**
> >
> > - **Thorough analyses of the hard negatives (Q3)**
> >
> > We would like to clarify that our paper systematically studies the concept of “hard negatives” and identifies their correlation with retriever strength. While we agree with the reviewer’s valid explanation of why this correlation exists, we emphasize that **our work is the first to observe and empirically verify this phenomenon**. The ability to provide a reasonable explanation for the correlation does not imply that the phenomenon itself is trivial or that it has been previously discovered. To the best of our knowledge, no prior work has systematically studied or validated this relationship, and our findings represent an important contribution to the field.
> >
> > Finally, we would also like to highlight our technical contributions under the sections “Novel methods for robust RAG” and “Comprehensive study of RAG-specific LLM tuning.” These contributions not only advance the understanding of long-context RAG but also offer practical solutions for improving system performance.
> >
> >
> > [1] Gemini 1.5: Unlocking multimodal understanding across millions of tokens of context
> >
> > [2] The Llama 3 Herd of Models
> >
> > [3] Mistral NeMo
> >
> > [4] Can Long-Context Language Models Subsume Retrieval, RAG, SQL, and More?
> >
> > [5] Post: https://x.com/Francis_YAO_/status/1758934303655030929, https://x.com/Francis_YAO_/status/1759962812229800012
> >
> > [6] Making Retrieval-Augmented Language Models Robust to Irrelevant Context
> >
> > [7] Chain-of-note: Enhancing robustness in retrieval-augmented language models

---

> > > ### Comment · Reviewer_ua3n · 2024-11-24
> > >
> > > Thank you for your reply. I have to admit that you are really good at telling stories. However, from the perspective of a senior RAG and information retrieval researcher, the actual technical contribution of your paper to the RAG community is still very limited. As other reviewers have pointed out, they also believe that your research is essentially about the robustness of RAG, which has been fully discussed in previous. As for the discussion between extremely long context and RAG that you mentioned, your work is between the two. It is neither a scenario of extremely long context nor an actual scenario of RAG. It has very limited practical applications in inspiring the next stage of RAG. Therefore, after carefully reading your paper and reply, I still feel that I cannot get enough inspiration for the RAG community (different from previous research) from this paper, so I cannot modify my review.

---

> > > > ### Author Response · Authors · 2024-11-24
> > > > **Round3 Author Response (1/2)**
> > > >
> > > > Thank you for taking the time to review our rebuttal and for sharing your feedback. We sincerely appreciate your perspective as a senior researcher in the RAG and information retrieval field. While we understand and respect your position, we would like to further clarify the key distinctions and contributions of our work to not only **RAG research** but also **long-context LLM research**, as well as address the concerns raised in your most recent response.
> > > >
> > > > - **Your work is neither a scenario of extremely long context nor an actual scenario of RAG.**
> > > >
> > > > As discussed in our previous response, there is an ongoing debate between (a) RAG with a small number of retrieved passages and (b) solely long-context LLMs. Our work, (c) RAG with a large number of retrieved passages, serves as a valuable intermediate step bridging (a) and (b). In this section, we would like to highlight the value of (c) to **(b) long-context LLMs**, leaving the discussion of (c) to (a) for the next question’s response.
> > > >
> > > > (1) **Strong evidence that (b) cannot currently replace (a)**: As shown in Figure 1, *simply filling the entire context window of long-context LLMs (8k/128k/2M tokens)* (represented by the rightmost point of each curve) does not guarantee the best performance. For all studied LLMs (except Gemini-1.5-pro), the performance of long-context RAG often fails to surpass that of RAG with a small number of retrieved passages, particularly when using a strong retriever (as illustrated in Figure 1(a)). This finding provides compelling evidence that (b) is not yet effective enough to replace (a) in practical scenarios.
> > > >
> > > > (2) **The need for real evaluations of (b)**: As discussed in Section 3.3, the *hard negatives* encountered in real-world RAG scenarios (will also be present if we fill the entire corpus into (b)) are significantly more challenging than random negatives. Moreover, we observe a correlation between retriever strength and the difficulty of these hard negatives. Existing benchmarks for evaluating long-context LLMs, such as Needle-in-the-Haystack [1] and RULER [2], primarily adopt *synthetic settings* with *random negatives* in the context. We argue that these synthetic evaluations are insufficient to assess the true capabilities of long-context LLMs in real RAG scenarios. Our findings underscore the need for more *realistic* evaluations that incorporate *hard negatives* in diverse and complex retrieval scenarios, especially if we aim for (b) to replace (a) in the future.
> > > >
> > > > (3) **Solutions to enhance (b)**: Given that (b) currently lacks the performance to replace (a), our work explores ways to enhance the capabilities of long-context LLMs. Specifically: We conduct data-augmented fine-tuning experiments to improve the performance of long-context LLMs, as described in Section 5. Our study investigates various fine-tuning design choices, such as data distribution, training context size, and other critical parameters, providing a practical "recipe" for future researchers aiming to strengthen (b) in real-world applications.

---

> > > > ### Author Response · Authors · 2024-12-01
> > > > **Kind Reminder from Authors**
> > > >
> > > > Thank you once again for your thoughtful feedback and engagement in the discussion phase. We deeply value your insights, which have significantly strengthened our submission.
> > > >
> > > > As the discussion phase **concludes in two days**, we wanted to highlight the clarifications and updates made in response to your feedback: (1) Clarified Motivation and Contribution: We have elaborated on how this work addresses the ongoing debate between Retrieval-Augmented Generation (RAG) and long-context LLMs. (2) Distinction from Prior Work: We have emphasized how this work differs from previous studies on “LLM robustness in RAG,” showcasing its unique contributions. (3) Added Comparisons: We have incorporated a comparison with additional existing works, as you suggested.
> > > >
> > > > We hope these updates address your concerns and demonstrate the significance of our contributions. It is also encouraging to note that **the motivation and contributions of this work have been acknowledged positively by other reviewers**.
> > > >
> > > > With these revisions in mind, we would be sincerely grateful **if you would consider reassessing your evaluation**. Please let us know if there are any remaining questions or points of discussion—we would be happy to continue the conversation before the deadline.

---

> > > > > ### Comment · Reviewer_ua3n · 2024-12-01
> > > > >
> > > > > After considering the author's rebuttal, revisions and other reviewers' feedback, I have revised my review. I have increased the score but decreased the confidence level. I hope to continue the discussion with AC and reviewers in the follow-up stage to decide my final review.

---

> > > > > ### Comment · Reviewer_ua3n · 2024-12-01
> > > > >
> > > > > I admit that in my initial comments, I had some misunderstandings about the contribution of this paper. After carefully reading the comments from other reviewers and the author's revised version, I have a new understanding of the contribution of this paper. As of this version, this paper makes unique contributions to previous work on RAG robustness. I understand the unique significance and contribution of this paper in the context of the rapid development of long-context LLM. So I have further improved my score and recommend acceptance of this paper.

---

> > > > > > ### Author Response · Authors · 2024-12-01
> > > > > > **Author Final Response**
> > > > > >
> > > > > > Thank you very much for your thoughtful and positive feedback! We are delighted to hear that our revisions have addressed your previous concerns. Your acknowledgment of our contributions and recommendation for acceptance means a lot to us. Wishing you all the best in your future endeavors!

---

> ### Author Response · Authors · 2024-11-22
> **Round2 Author Response (1/2)**
>
> Before answering your follow-up questions, we would like first to clarify the motivation behind our work and show that the problem researched in this paper (long-context RAG problem) is valid.
>
> With the rapid advancement of powerful long-context LLMs [1][2][3], there is an ongoing debate about whether these models can fully replace retrieval-augmented generation (RAG) systems that rely on a small number of retrieved passages in real-world applications [4][5]. A prevailing argument suggests that, given a sufficiently capable long-context LLM, the intermediate retrieval step could be eliminated by directly inputting the entire knowledge corpus into the model’s context [4][5]. However, the transition from (a) RAG with a small number of retrieved passages to (b) a system solely reliant on long-context LLMs introduces an important intermediate stage: (c) RAG with a large number of retrieved passages. This intermediate stage (c) can be seen as an extension of (a) and a subset of (b). If long-context LLMs (b) can genuinely replace RAG systems (a), then the performance of (c) should at least match or even surpass that of (a).
>
> Motivated by this insight, our work investigates (c)—RAG with long-context LLMs leveraging a large number of retrieved passages—and systematically evaluates whether (b) can truly replace (a). Furthermore, we propose solutions to improve the robustness and performance of long-context LLMs for RAG applications. By addressing this intermediate stage, our work sheds light on a critical aspect of the transition between retrieval-augmented systems and fully context-based LLMs, which has not been adequately explored in existing literature.
>
> Then, we would like to answer your following questions:
>
> - **You have essentially taken a problem that has already been well-studied in previous work. Fundamentally a matter of RAG robustness. (Q1, Q2)**
>
> We agree that, in a broader sense, our work is related to improving the robustness of LLM generation with noise input. However, we would like to point out the distinction of our work: (1) *Discovery of U-curve in long-context RAG*: even though the reviewer provides a valid explanation of the U-curve phenomenon, **this phenomenon is not shown in existing works [6][7] with short-context RAG (< 5 retrieved passages)**. It can only be observed with long-context RAG. (2) *Systematic analysis of how different retrievers and LLMs affect the noise robustness*: We conduct a systematic study examining how RAG performance varies across **different LLMs and retrievers**. For example, before our work, it was not known that certain strong LLMs, such as Gemini, demonstrate remarkable robustness to hard negatives and show continuously improving performance as the number of retrieved passages grows. In contrast, other open-source LLMs, such as Gemma and Mistral, exhibit significant sensitivity to hard negatives, resulting in an inverse U-shaped performance curve. Our work provides empirical evidence to support these observations and establishes a foundation for further analysis **while existing works [6][7] mainly focus on either Llama-2 7B or 13B**. (3) *Serious analyses on hard negatives*: While previous studies have identified the existence of hard negatives (as we have cited), we go beyond this by showing that **precision alone is insufficient to quantify their difficulty**. We also investigate their correlation with retriever strength, providing deeper insights into the interplay between retrieval quality and LLM robustness, which is not shown in existing works. (4) *Comprehensive RAG-specific finetuning study*: We perform a comprehensive study of the various design components of RAG-specific LLM tuning (presented in Sections 5,6 and Appendices M and N), to provide design hints for researchers who are interested in implementing RAG-specific LLM tuning for their work or research. This is not studied in existing works.

---

> ### Author Response · Authors · 2024-11-24
> **Round3 Author Response (2/2)**
>
> - **Your paper's actual technical contribution to the RAG community is still minimal.**
>
> We appreciate the reviewer’s feedback and would like to clarify the value of our work to **(a) the RAG community**, as well as the distinct technical contributions we make.
>
> (1) **Empirical study of the dynamics between retriever and LLM generator**: In an RAG system, there is an inherent balance between the retriever and the LLM generator. A strong retriever can retrieve relevant information in a small top-k set, reducing the need for a highly robust LLM. Conversely, a weak retriever requires a larger top-k set to find useful passages, necessitating a stronger and more robust long-context LLM. While this relationship might seem intuitive, we are the first to **empirically demonstrate these dynamics** between different retrievers and LLM generators, as detailed in Section 3.1. This insight provides valuable guidance for optimizing the interaction between retrieval and generation components in RAG systems.
>
> (2) **Deep understanding of hard negatives**. We acknowledge the reviewer’s broader observation that our work relates to improving the robustness of LLMs against noisy inputs. However, our contribution goes beyond prior research by offering a **systematic analysis of hard negatives**: We show that **precision alone is insufficient to quantify the difficulty of hard negatives**, emphasizing the need for a nuanced evaluation of retrieval quality. We analyze the **correlation between retriever strength and the difficulty of hard negatives**, providing deeper insights into the interplay between retrieval quality and LLM robustness. This level of analysis has not been addressed in existing works and offers actionable insights for designing more resilient RAG systems.
>
> (3) **Retrieval reordering**: In scenarios where a strong retriever is unavailable (e.g., out-of-distribution domains for pretrained dense retrievers), increasing the number of retrieved passages is often necessary to achieve acceptable recall. In such cases, we introduce **retrieval reordering** (Section 4) as a novel solution. This method is simple, **plug-and-play**, and highly efficient; demonstrates significant performance improvements in challenging retrieval scenarios; and represents an **effective strategy to bridge the gap between retrieval and generation components** when dealing with suboptimal retrieval quality.
>
> (4) **Comprehensive RAG-specific finetuning study**: Our work also conducts a **comprehensive study of the various design components involved in RAG-specific LLM tuning**, as presented in Sections 5, 6, and Appendices M and N. These include key factors such as data distribution, training context size, and evaluation settings, which provide insights and practical guidelines for researchers and practitioners aiming to implement RAG-specific LLM tuning in their systems. This study fills a gap in the current literature, providing valuable design hints and actionable recommendations for advancing RAG-specific LLM tuning.
>
>
> - **As other reviewers have pointed out, they also believe that your research is essentially about the robustness of RAG.**
>
> We appreciate the reviewer’s perspective and agree that, in a broader sense, our work contributes to improving the robustness of LLM generation against noisy inputs. However, as outlined in our previous rebuttal, our research distinguishes itself in several key ways, **which have also been acknowledged by other reviewers**: (1) Empirical study of the dynamics between retriever and LLM generator; (2) Deep understanding of hard negatives; (3) Systematic analysis of noise robustness across retrievers and LLMs; (4) Comprehensive RAG-specific finetuning study.
>
> We would appreciate it if the reviewer could re-evaluate the contribution of the paper from both the long-context LLM perspective and the RAG perspective. We thank the reviewer again for your valuable feedback and welcome any further suggestions to strengthen our work.
>
> [1] https://github.com/gkamradt/LLMTest_NeedleInAHaystack
>
> [2] RULER: What's the Real Context Size of Your Long-Context Language Models?

---

### Official Review · Reviewer_k49Q · 2024-10-30

**Soundness:** 3
**Presentation:** 4
**Contribution:** 3
**Rating:** 6
**Confidence:** 4

**Summary:**

This paper primarily explores the phenomenon of long inputs in Retrieval-Augmented Generation (RAG): for many long-context large language models (LLMs), the quality of generated output initially improves but then declines as the number of retrieved passages increases. This observation contradicts the previous assumption that a larger retrieval set, containing more relevant information, would enhance performance. The study identifies the detrimental impact of retrieved "hard negatives" as a significant factor. To mitigate this effect and enhance the robustness of long-context LLMs in RAG, the paper proposes three methods: (1) an untrained method based on retrieval reordering, (2) implicit robustness fine-tuning to enhance resistance to hard negatives, and (3) explicit relevance fine-tuning with intermediate reasoning to identify relevance.

**Strengths:**

1. The phenomenon where the quality of LLM-generated outputs first improves and then declines with an increasing number of retrieved segments is intriguing, and the authors have conducted thorough experiments to explore this.
2. The importance of hard negatives for long-context LLM evaluation is well-studied, and it is observed that increasing the number of hard negatives typically leads to a decrease in RAG answer accuracy. The strength of the retriever is directly related to the difficulty of the hard negatives.
3. The three methods proposed by the authors (Retrieval reordering, Implicit robustness fine-tuning, Explicit relevance fine-tuning) all enhance the performance of long-context LLMs in RAG applications.
4. Overall, the paper is well-written and easy to follow.

**Weaknesses:**

1. Figure 1 shows that while other LLMs exhibit an initial improvement in performance followed by a decline with an increasing number of retrieved passages, Gemini-1.5-Pro does not exhibit this trend, and the authors do not explain this phenomenon.
2. From the experiments, it appears that LLMs ranging from 7B to 12B are susceptible to the influence of hard negatives. It remains unclear whether this phenomenon can still be observed in LLMs larger than 30B. I suspect that larger LLMs might automatically identify these hard negatives.

**Questions:**

See Questions

---

> ### Author Response · Authors · 2024-11-20
> **Author Response (1/1)**
>
> We appreciate you taking the time to provide such thorough and insightful feedback! We address your comments below:
>
> - **The reason why Gemini-1.5-Pro does not exhibit the increase-then-decrease trend.**
>
> We hypothesize that the robustness of Gemini-1.5-Pro to hard negatives stems from its advanced training methodology and high-quality pretraining/post-training data. To further investigate the factors influencing LLM robustness, we designed experiments to analyze the impact of: (1) context window length, (2) model size, and (3) pretraining and post-training data.
>
> Focusing on the first two factors, we conducted experiments on the NQ dataset using the e5 retriever and two groups of LLMs with varying model sizes and context window lengths: (1) 8K window LLMs: llama3-8B-Chat, gemma-2-9B-Chat and llama3-70B-Chat; (2) 128K window LLMs: llama3.1-8B-Chat, mistral-nemo-12B-Instruct and llama3.1-70B-Chat.
>
>
> | # retrieved psg  | 1  | 5  | 10 | 20  | 30 | 40 | 50 | 100 | 150 | 500 |
> |----|---|----|---|----|---|---|---|---|---|---|
> | llama3-8B-Chat (8k)| 0.4432 | 0.5166 | 0.5161 | **0.5199** | 0.5025 | 0.4681 | -  | - | - | - |
> | gemma-2-9B-Chat (8k) | 0.4687 | 0.5296 | **0.5435** | 0.5421 | 0.5271 | 0.5166 | - | - | - | - |
> | llama3-70B-Chat (8k) | 0.4454 | 0.5360 | 0.5576 | 0.5742 | **0.5839** | 0.5706 | - | - | - | - |
> | llama3.1-8B-Chat (128k) | 0.4684 | 0.5410 | **0.5474** | 0.5429 | 0.5355 | 0.5346 | 0.5343 | 0.5066 | 0.5025 | 0.4288 |
> | mistral-nemo-12B-Instruct (128k) |  0.4598 | 0.5355 | 0.5393 | **0.5593** | 0.5452 | 0.5474 | 0.5255 | 0.4825 |  0.4366 | 0.2856 |
> | llama3.1-70B-Chat (128k) | 0.4789 | 0.5490 | **0.5607** | 0.5449 | 0.5429 | 0.5343 | 0.5427 | 0.5197 | 0.5186 | 0.4665 |
>
>
> Our findings indicate **no clear correlation between the context window size and robustness in increasing passage numbers.** Contrary to expectations, the smaller context LLM (llama3-70B-chat with 8k) exhibited greater robustness than the larger context LLM (llama3.1-70B-chat with 128k). Furthermore, models with the same maximum context window length (e.g., llama3.1-8B-chat and Mistral-Nemo-12B-Instruct) demonstrated significantly different robustness levels.
>
> These results suggest that factors beyond context window size, such as pretraining or post-training data, may play a crucial role in determining LLM robustness.  We hypothesize that despite supporting larger context windows, LLMs might exhibit a bias towards shorter contexts due to the prevalence of shorter training samples.  However, without access to the specific training data used for these LLMs, further investigation is needed to confirm this hypothesis. We acknowledge this limitation and recommend it as an area for future research.
>
>
> - **If LLMs larger than 30B are susceptible to hard negatives.**
>
> Thank you for raising this important question. To investigate whether LLMs larger than 30B parameters exhibit similar susceptibility to hard negatives, we conducted experiments with Llama-3.1-70B-Chat on the NQ dataset using both e5 and BM25 retrievers.
>
> | # retrieved psg | 1 | 5 | 10  | 20  | 30  | 40 | 50 | 100 | 150 | 500 |
> |---|--|--|--|----|---|---|--|---|---|--|
> | llama3.1-70B-Chat w. e5 | 0.4789 | 0.5490 | **0.5607** | 0.5449 | 0.5429 | 0.5343 | 0.5427 | 0.5197  | 0.5186 | 0.4665 |
> | llama3.1-70B-Chat w. BM25 | 0.2637 | 0.3659 | 0.4249 | 0.4532 | **0.4809** | 0.4778 | 0.4773 | 0.4759 | 0.4695 | 0.4485 |
>
>
> The results clearly demonstrate that even this larger LLM remains sensitive to hard negatives.  The characteristic inverted U-shaped performance curve persists as the number of retrieved passages increases. This suggests that model size alone is not the determining factor in LLM robustness to hard negatives.
>
> We suspect that the quality and nature of pretraining and post-training data play a more significant role. However, due to the lack of publicly available information regarding the training data used for these LLMs, we are unable to definitively confirm this hypothesis at present. We acknowledge this limitation and recommend further investigation into the role of training data as a promising avenue for future research.

---

> ### Comment · Reviewer_k49Q · 2024-11-22
>
> Thank you for your response. Upon reevaluating the revised paper, I have identified some additional concerns. The paper primarily explores how, for many long-context LLMs, the quality of the generated output initially improves but then subsequently declines as the number of retrieved passages increases. However, it seems there might be more rational ways to address this:
>
> 1. The "hard negatives" phenomenon is caused by inaccuracies in the retriever. A simpler approach could be to enhance the quality of the Retriever. If the quality of the retrieval results is high enough, further denoising by the LLM may not be necessary, although the robustness of the LLM is still worth studying.
> 2. In terms of the number of retrieval passages, RAG does not need to retrieve such long passages in practical use.
>
> Based on the two considerations mentioned above, I decide to lower my score from 8 to 6.

---

> > ### Author Response · Authors · 2024-11-22
> > **Round2 Author Response (1/1)**
> >
> > Before answering your follow-up questions, we would like first to clarify the motivation behind our work and show that the problem researched in this paper (long-context RAG problem) is valid and important.
> >
> > With the rapid advancement of powerful long-context LLMs [1][2][3], there is an ongoing debate about whether long-context LLMs can fully replace retrieval-augmented generation (RAG) systems that use a small number of retrieved passages for real-world applications [4][5]. The prevailing argument is that given a sufficiently capable long-context LLM, one could bypass the intermediate retrieval step entirely by integrating the entire knowledge corpus into the model’s context. However, the transition from (a) RAG with a small number of retrieved passages to (b) solely relying on long-context LLMs introduces an intermediate stage: (c) RAG with a large number of retrieved passages. This intermediate stage (c) can be viewed as an extension of (a) and a subset of (b). If long-context LLMs (b) are truly capable of replacing RAG systems (a), then the performance of (c) should closely align with or even surpass that of (a). Motivated by this insight, our work investigates (c)—RAG with long-context LLMs using a large number of retrieved passages—and analyzes whether (b) can genuinely replace (a). Furthermore, we propose solutions to enhance long-context LLMs for RAG applications.
> >
> > Now, we would like to answer your specific questions:
> >
> > - **If the quality of the retrieval results is high enough, further denoising by the LLM may not be necessary, although the robustness of the LLM is still worth studying. (Q1)**
> >
> > We agree with the reviewer that research on the retriever side is indeed crucial, as enhancing the retriever’s capabilities would undoubtedly improve the overall performance of RAG systems. However, it is important to acknowledge that **no retriever is perfect**; challenges such as *hard negatives* will always persist. Addressing these issues requires complementary efforts on the LLM side. Improving the robustness of LLMs to such hard negatives, or enabling them to effectively identify and mitigate the impact of these negatives, is equally important. As demonstrated in this paper, these advancements can significantly enhance the performance of the entire RAG system.
> >
> > - **In terms of the number of retrieval passages, RAG does not need to retrieve such long passages in practical use. (Q2)**
> >
> > We would like to answer this comment from two perspectives: **(1) Our motivation**: As claimed in the second paragraph of this rebuttal (our motivation paragraph), the problem we approached in this paper (c) can be seen as an intermediate step of the transition from (a) to (b). We conduct the research on (c) - RAG with a large number of retrieved passages, in order to have a conclusion and end the debate of whether (b) can really replace (a) for the current stage and propose solutions to enhance long-context LLMs for RAG applications on the way from (a) to (b). **(2) Increasing the number of retrieved passages helps improve the RAG performance**: As we can see in Figure 1, with the BM25 retriever, Gemma, Gemma2 and Mistral reach the peak RAG performance with 30 or 50 retrieved passages (much larger than “< 5 retrieved passages” adopted in existing works [6][7][8]); with the e5 retriever, Mistral and Gemini-1.5-pro reach the performance peak at 20 and 500 respectively. These results clearly indicate that retrieving many passages can still be beneficial in practice as well.
> >
> > [1] Gemini 1.5: Unlocking multimodal understanding across millions of tokens of context
> >
> > [2] The Llama 3 Herd of Models
> >
> > [3] Mistral NeMo
> >
> > [4] Can Long-Context Language Models Subsume Retrieval, RAG, SQL, and More?
> >
> > [5] Post: https://x.com/Francis_YAO_/status/1758934303655030929, https://x.com/Francis_YAO_/status/1759962812229800012
> >
> > [6] Making Retrieval-Augmented Language Models Robust to Irrelevant Context. ICLR 2024.
> >
> > [7] RA-DIT: Retrieval-Augmented Dual Instruction Tuning. ICLR 2024.
> >
> > [8] Self-RAG: Learning to Retrieve, Generate, and Critique through Self-Reflection. ICLR 2024

---

> > > ### Comment · Reviewer_k49Q · 2024-11-23
> > >
> > > Thank you for your prompt response. I have no further questions. Considering the contributions of the paper, I think Rating 6 is appropriate.

---

> > > > ### Author Response · Authors · 2024-11-24
> > > > **Round3 Author Response (1/1)**
> > > >
> > > > Thank you for your thoughtful feedback and for acknowledging our responses!
> > > >
> > > > We are pleased that we were able to address your concerns and clarify the motivation and contributions of our work. We would greatly appreciate it if you could share which specific aspects of our contributions or any remaining areas of confusion led to the score change from 8 to 6. We would be more than happy to continue the discussion and provide further clarifications or adjustments before the deadline to ensure that all concerns are fully addressed.

---

### Official Review · Reviewer_vMLV · 2024-11-03

**Soundness:** 2
**Presentation:** 3
**Contribution:** 2
**Rating:** 6
**Confidence:** 3

**Summary:**

This paper investigates the challenges and opportunities of using long-context LLMs in RAG systems. The main contributions include:
- A systematic analysis demonstrating that increasing retrieved passages doesn't always improve performance with long-context LLMs
- Identification of the detrimental impact of "hard negatives" on LLM performance
- Three approaches to improve RAG performance: (1) a training-free retrieval reordering method, (2) Implicit robustness fine-tuning, (3) explicit relevance fine-tuning with intermediate reasoning.
- Comprehensive analysis of design choices for RAG-specific LLM tuning.

**Strengths:**

- The paper presents a technically sophisticated analysis of long-context LLMs in RAG systems, with comprehensive experiments across multiple models, retrievers, and datasets. The empirical validation is thorough and well-documented.
- The identification of the "hard negatives" problem in long-context RAG represents a major contribution to understanding LLM behavior in retrieval contexts.
- The proposed methods, particularly the training-free retrieval reordering approach, offer immediate practical value for improving RAG systems. The solutions are well-motivated by the empirical findings.

**Weaknesses:**

- The paper lacks to explain why long-context LLMs behave differently with hard negatives, which may limit the generalizability of the findings and makes it harder to predict when the proposed solutions might fail.
- The experimental evaluation doesn't compare against recent competing methods for improving RAG performance, such as reranking approaches or hierarchical retrieval methods.
- The paper lacks an analysis of failure cases or limitations. When do these methods break down? What types of queries or documents are particularly challenging?
- The computational requirements of the intermediate reasoning approach requiring Gemini-1.5-Pro are not thoroughly discussed. This raises questions about the practical applicability in resource-constrained environments.

**Questions:**

- What are the key properties of "hard negatives" that make them particularly challenging? Is it semantic similarity, structural patterns, or something else? How does your analysis generalize to other types of retrievers not tested in the paper?
- How does your approach compare to recent methods like RetRobust (Yoran et al. 2023 - Making Retrieval-Augmented Language Models Robust to Irrelevant Context), or RA-DIT: Retrieval-Augmented Dual Instruction Tuning (Lin et al. 2023) that also aim to improve RAG robustness?
- Have you considered comparing against hierarchical retrieval methods that might naturally mitigate the hard negatives problem?
- Could you provide a cost-benefit analysis comparing your methods against simpler approaches like reducing retrieval size or basic reranking?
- Are there specific types of queries or documents where hard negatives are particularly problematic?
- Why did you choose the specific reordering strategy for the training-free approach? Have you explored other ordering patterns?
- How sensitive is the intermediate reasoning approach to the quality and style of reasoning labels?

---

> ### Author Response · Authors · 2024-11-20
> **Author Response (1/5)**
>
> We appreciate your thorough review and insightful comments! To ensure clarity and conciseness, we have grouped similar points raised in the weakness and question sections and addressed them as follows.
>
> - **Why long-context LLMs behave differently with hard negatives (W1).**
>
> Thank you for your insightful question. We believe the observed differences in how long-context LLMs handle hard negatives could be attributed to three primary factors: (1) context window length, (2) model size, and (3) pretraining and post-training data.
>
> To disentangle the influence of context window length and model size, we conducted experiments on the NQ dataset using the e5 retriever and two groups of LLMs with varying model sizes: (1) 8K window LLMs: llama3-8B-Chat, gemma-2-9B-Chat and llama3-70B-Chat; (2) 128K window LLMs: llama3.1-8B-Chat, mistral-nemo-12B-Instruct and llama3.1-70B-Chat.
>
>
> | # retrieved psg  | 1 | 5 | 10  | 20 | 30 | 40 | 50 | 100 | 150 | 500 |
> |---|---|---|--|----|----|--|---|---|--|--|
> | llama3-8B-Chat (8k) | 0.4432 | 0.5166 | 0.5161 | **0.5199** | 0.5025 | 0.4681 | -  | - | - | - |
> | gemma-2-9B-Chat (8k)| 0.4687 | 0.5296 | **0.5435** | 0.5421 | 0.5271 | 0.5166 | - | - | - | - |
> | llama3-70B-Chat (8k) | 0.4454 | 0.5360 | 0.5576 | 0.5742 | **0.5839** | 0.5706 | -  | - | - | - |
> | llama3.1-8B-Chat (128k) | 0.4684 | 0.5410 | **0.5474** | 0.5429 | 0.5355 | 0.5346 | 0.5343 | 0.5066 | 0.5025 | 0.4288 |
> | mistral-nemo-12B-Instruct (128k) |  0.4598 | 0.5355 | 0.5393 | **0.5593** | 0.5452 | 0.5474 | 0.5255 | 0.4825 |  0.4366 | 0.2856 |
> | llama3.1-70B-Chat (128k)  | 0.4789 | 0.5490 | **0.5607** | 0.5449 | 0.5429 | 0.5343 | 0.5427 | 0.5197  | 0.5186 | 0.4665 |
>
> Our findings indicate **no clear correlation between the context window size and robustness in increasing passage numbers.** Contrary to expectations, the smaller context LLM (llama3-70B-chat with 8k) exhibited greater robustness than the larger context LLM (llama3.1-70B-chat with 128k). Furthermore, models with the same maximum context window length (e.g., llama3.1-8B-chat and Mistral-Nemo-12B-Instruct) demonstrated significantly different robustness levels.
>
> These results suggest that factors beyond context window size, such as pretraining or post-training data, may play a crucial role in determining LLM robustness.  We hypothesize that despite supporting larger context windows, LLMs might exhibit a bias towards shorter contexts due to the prevalence of shorter training samples.  However, without access to the specific training data used for these LLMs, further investigation is needed to confirm this hypothesis. We acknowledge this limitation and recommend it as an area for future research.

---

> ### Author Response · Authors · 2024-11-20
> **Author Response (2/5)**
>
> - **Comparison with reranking approaches or hierarchical retrieval methods (W2, Q3).**
>
> Thank you for this valuable suggestion.  Our focus in this paper is on analyzing and improving the performance of long-context LLMs as **generators** in RAG systems. While reranking and hierarchical retrieval are important techniques for enhancing the **retriever** component, they fall outside the primary scope of our work.
>
> However, we recognize the value of exploring the interplay between our proposed methods and advanced retrieval techniques. To address your comment, we have included additional experiments on PopQA and TriviaQA incorporating a hierarchical retrieval component (referred to as reranking).  Specifically, we first retrieve 200 passages using the e5 retriever and then apply reranking with bge-reranker-large [1].  Llama3-8B-chat serves as the generator in these experiments.
>
>
> PopQA:
>
> | # retrieved psg | 5 | 10 | 20 | 30 | 40 |
> |---|---|---|---|--|--|
> | RAG |  0.5039 | 0.5069 |  0.4988 | 0.4867 | 0.4317 |
> | RAG w. reordering (Ours)  | 0.5076 | 0.5076 | 0.5056 | 0.5049 | 0.4808 |
> | RAG w. reranking  | 0.5084 | 0.5232 |  0.5211 |  0.5031 | 0.4564 |
> | RAG w. reranking+reordering (Ours) | 0.5050 | 0.5217 | 0.5267 | 0.5197 |  0.4988 |
> | RAG w. implicit tuning (Ours) | 0.5524 | 0.5599 | 0.5606 | 0.5545 | 0.5574 |
> | RAG w. intermediate reasoning tuning (Ours) | 0.6102 | 0.6207 | 0.6292 | 0.6319 | 0.6244 |
>
>
> TriviaQA:
>
> | # retrieved psg | 5 | 10  | 20 | 30 | 40 |
> |---|--|---|---|--|--|
> | RAG | 0.6833 | 0.7006 |  0.7247 | 0.7264 | 0.7107 |
> | RAG w. reordering (Ours) | 0.6835 | 0.7022 |   0.7262 | 0.7249 | 0.7182 |
> | RAG w. reranking | 0.7009 | 0.7219 | 0.7390 | 0.7362 | 0.7223 |
> | RAG w. reranking+reordering (Ours) | 0.6984 | 0.7200 | 0.7356 | 0.7399 | 0.7340 |
> | RAG w. implicit tuning (Ours) | 0.7277 | 0.7399 |  0.7459 | 0.7515 |  0.7542 |
> | RAG w. intermediate reasoning tuning (Ours) | 0.7734 | 0.7861 | 0.7959 | 0.7983 |  0.7995 |
>
> The results demonstrate two key findings: (1) *Superiority of our proposed methods*: Both RAG-specific implicit tuning and RAG tuning with intermediate reasoning significantly outperform the baseline RAG system with reranking. (2) *Complementary benefits*: Combining reordering with reranking (RAG w. reranking+reordering) further enhances performance, especially when retrieving a large number of passages, compared to reranking alone (RAG w. reranking). This analysis highlights the effectiveness of our proposed methods even in conjunction with advanced retrieval techniques like reranking.

---

> ### Author Response · Authors · 2024-11-20
> **Author Response (3/5)**
>
> - **The cost-benefit analysis compared with reducing retrieval size and basic reranking (Q4).**
>
> To provide a comprehensive assessment, we analyze the cost-benefit trade-offs between basic reranking, retrieval reordering (Ours), and RAG-specific LLM tuning (Ours).
>
> **Basic reranking.** *Cost*: Reranking involves processing every query in each testing dataset with a computationally expensive cross-encoder model. This process can be time-consuming, especially for large datasets. In our experiments, reranking the top 200 passages on the TriviaQA dataset (over 11k samples) took over 1.5 hours (with bge-reranker-large [1]) using DataParallel and 8 H100s (while retrieval only (without reranking) with e5 and gpu-based approximated nearest neighbor search - faiss-gpu - only takes 10 mins). This cost scales linearly with dataset size, making it a significant bottleneck for large-scale applications. *Benefit*: Reranking can improve retrieval accuracy, leading to better overall RAG performance, as demonstrated in our results above.
>
> **Retrieval reordering (Ours).** *Cost*: This method is designed to be plug-and-play and can be seamlessly integrated with any existing RAG pipeline (retriever, reranker, and generator) without incurring additional computational costs; *Benefit*: It consistently improves RAG performance, particularly when using a larger number of retrieved passages. Importantly, it is both retriever-agnostic and reranker-agnostic, demonstrating its versatility and broad applicability (as shown in Figure 4 and the table above).
>
> **RAG-specific LLM tuning (Ours).** *Cost*: Training an LLM requires computational resources and time. In our experiments, achieving a well-trained checkpoint took approximately 2 hours using DeepSpeed zero3 and 8 H100s. *Benefit*: This method yields substantial and consistent improvements in the LLM's ability to handle RAG scenarios across different numbers of retrieved passages (as shown in the table above). Crucially, this training is a one-time cost; the resulting LLM can be deployed for any testing dataset without incurring further computational overhead during inference (retrieval with e5 and gpu-based approximated nearest neighbor search - faiss-gpu - only takes 10 mins on TriviaQA, while reranking with bge-reranker-large [1] takes more than 1.5 hours).
>
> It's important to highlight that these three approaches are orthogonal and address different aspects of the RAG system: **Basic reranking** focuses on improving the quality of retrieved passages. **Retrieval reordering** optimizes the interaction between the retriever and the LLM generator. **RAG-specific LLM tuning** enhances the generator's capabilities for handling retrieved information. This inherent complementarity suggests that combining these techniques can lead to further performance gains in RAG systems.
>
>
> - **Lack of analysis of failure cases. When do these methods break down? What types of queries are challenging with hard negatives?  (W3, Q5)**
>
> We provide detailed case studies in the appendices to further illustrate our findings: *Appendix D*: Examines retrieved hard negatives, *Appendix I*: Demonstrates the effectiveness of our proposed methods, *Appendix O (revised manuscript)*: Presents new case studies focusing on the analysis of challenging queries.
>
> Our analysis of these challenging queries reveals **no easily discernible pattern** that distinguishes those queries which are answerable with a small number of retrieved passages but fail with larger retrieval sets. We hypothesize that these failures stem from complex interactions between the query, the retriever, and the knowledge corpus.  Specifically, if increasing the retrieval top-k primarily introduces noisy or irrelevant information, while the relevant information was already present in a smaller top-k set, the likelihood of the LLM failing to answer correctly increases significantly.

---

> ### Author Response · Authors · 2024-11-20
> **Author Response (4/5)**
>
> - **The computational requirements of the intermediate reasoning approach with Gemini-1.5-Pro. How sensitive is the intermediate reasoning approach to the quality and style of reasoning labels? (W4, Q7)**
>
> You raise a valid concern about the cost of generating reasoning labels using a large language model like Gemini-1.5-pro, especially for large datasets. To address this, we conducted additional experiments where we fine-tuned the LLM using reasoning labels generated by a smaller and more cost-effective model, llama3-8B-chat. This allows us to assess the robustness of our proposed method to variations in the quality of reasoning labels.
>
> TriviaQA
>
> | # retrieved psg  | 5 | 10 | 20 | 30 | 40 |
> |----|---|----|---|---|---|
> | RAG  | 0.6833 | 0.7006 |  0.7247 | 0.7264 | 0.7107 |
> | RAG implicit tuning  | 0.7277 | 0.7399 |  0.7459 | 0.7515 |  0.7542 |
> | RAG tuning w. reasoning (llama3-8B-chat generated) | 0.7532 | 0.7700 | 0.7728 |  0.7708 | 0.7515 |
> | RAG tuning w. reasoning (Gemini-1.5-pro generated) | 0.7734 | 0.7861 | 0.7959 | 0.7983 |  0.7995 |
>
> PopQA
>
> | # retrieved psg  | 5 | 10 | 20 | 30  | 40 |
> |---|---|--|--|--|---|
> | RAG |  0.5039 | 0.5069 |  0.4988 | 0.4867 | 0.4317 |
> | RAG implicit tuning  | 0.5524 | 0.5599 | 0.5606 | 0.5545 | 0.5574 |
> | RAG tuning w. reasoning (llama3-8B-chat generated) | 0.6125 | 0.6200 | 0.6172 | 0.5985 | 0.5760 |
> | RAG tuning w. reasoning (Gemini-1.5-pro generated) | 0.6102 | 0.6207 | 0.6292 | 0.6319 | 0.6244 |
>
> The results demonstrate the resilience of our approach. While using labels from a powerful LLM like Gemini-1.5-pro yields the best performance, utilizing labels from a smaller LLM like llama3-8B-chat still delivers significant improvements.  Specifically, it outperforms both RAG implicit tuning and the vanilla RAG baseline. This finding offers a valuable practical trade-off: users can opt for smaller, less expensive LLMs for generating reasoning labels depending on their budget constraints, while still achieving substantial performance gains with our proposed method.
>
>
> - **What are the key properties of "hard negatives" which make them particularly challenging? Is it semantic similarity, structural patterns, or something else? How does your analysis generalize to other types of retrievers not tested in the paper? (Q1)**
>
> To further illustrate the nature of hard negatives, we present detailed case studies in Appendix D. Our analysis reveals that many hard negatives are passages that exhibit some relevance to the question but ultimately do not contain the correct answer. These passages often mention entities present in the question, leading to a degree of semantic similarity, but they fail to provide the necessary information to answer it accurately.
>
> Furthermore, to demonstrate the generalizability of the observed "inverted U-curve" phenomenon, we expanded our experiments on the NQ dataset to include additional retrievers beyond e5 and BM25, namely contriever and bge.  Our findings consistently show the presence of the inverted U-curve across all these retrievers.
>
>
> | # retrieved psg | 0  | 1  | 5 | 10 | 20 | 30 | 40 |
> |----|---|---|---|---|---|--|--|
> | llama3-8B-Chat w. BM25 | 0.2529 | 0.2130 | 0.3188 | 0.3670 | 0.4169 |  **0.4241** | 0.4202 |
> | llama3-8B-Chat w. contriever | 0.2529 | 0.2427 | 0.3654 | 0.4039 | 0.4368 | **0.4582** | 0.4440 |
> | llama3-8B-Chat w. bge | 0.2529 | 0.4042 | 0.4881 | **0.5061** | **0.5055** | 0.4975 |  0.4634 |
> | llama3-8B-Chat w. e5 | 0.2529 | 0.4432 | **0.5166** |  **0.5161** | **0.5199** | 0.5025 | 0.4681 |

---

> ### Author Response · Authors · 2024-11-20
> **Author Response (5/5)**
>
> - **Comparison with RetRobust and RA-DIT (Q2).**
>
> Thank you for highlighting these relevant models. While both RetRobust and RA-DIT aim to improve LLM robustness in RAG contexts, they differ from our work in key aspects.
>
> RetRobust focuses on mitigating the impact of irrelevant retrieved passages by employing strategies at the *retrieval-level*, primarily deciding whether to use retrieved information or not. In contrast, our approach delves deeper into *passage-level* relevance within long-context scenarios, aiming to equip the LLM with a finer-grained understanding of the retrieved information.
>
> RA-DIT, on the other hand, explores dual training of the retriever and LLM but primarily focuses on scenarios with small retrieved contexts (less than 10 passages). Unfortunately, due to the lack of publicly available code and resources for RA-DIT, we were unable to include it in our comparison.
>
> Therefore, we focused our comparative analysis on RetRobust, evaluating both its NLI strategy and LLM tuning strategy against our proposed methods on the NQ and PopQA datasets (retriever: e5, LLM: llama3-8B-chat).
>
> NQ
>
> | # retrieved psg | 1 | 5 | 10 | 20 |
> |---|----|---|--|---|
> | RetRobust (NLI) | 0.3864 | 0.3485 | 0.4380 | 0.4573 |
> | RetRobust (tuning)  | 0.4781 | 0.5042 | 0.5017 | 0.4814 |
> | RAG implicit tuning (Ours) | 0.4983 | 0.5640 | 0.5903 | 0.6003 |
> | RAG tuning with reasoning (Ours) | 0.5673 | 0.6659 | 0.6792 | 0.6928 |
>
> PopQA
>
> | # retrieved psg | 1 | 5 | 10 | 20 |
> |---|---|----|--|---|
> | RetRobust (NLI) | 0.2641 | 0.2603 | 0.4063 | 0.4197 |
> | RetRobust (tuning) |  0.4727 | 0.5166 | 0.5027 | 0.4544 |
> | RAG implicit tuning (Ours) | 0.4820 | 0.5524 | 0.5599 | 0.5606 |
> | RAG tuning with reasoning (Ours) | 0.5234 | 0.6102 | 0.6207 | 0.6292 |
>
> As the results demonstrate, our proposed RAG implicit tuning and RAG tuning with intermediate reasoning consistently and significantly outperform both RetRobust strategies. This highlights the effectiveness of our methods in enhancing LLM robustness to hard negatives, particularly in long-context RAG scenarios where the LLM needs to process and synthesize information from a larger number of retrieved passages.
>
>
> - **The choice of the reordering strategies (Q6).**
>
> You correctly identify the motivation behind our proposed retrieval reordering strategy: to counteract the "lost-in-the-middle" phenomenon inherent in LLMs. By strategically placing high-scoring (and thus, more likely relevant) retrieved passages at the beginning and end of the input sequence, we aim to capitalize on the LLMs' tendency to prioritize information in these positions.
>
> While we acknowledge the potential of exploring alternative reordering patterns, our primary goal in this paper is to establish the significance of passage ordering in RAG systems and introduce the concept of "position engineering."  Given the broader scope of our work, which encompasses a comprehensive analysis of long-context LLMs in RAG and the development of multiple solutions, an exhaustive investigation of various reordering strategies falls outside the current focus. We consider this a promising direction for future research and appreciate your insightful suggestion.
>
> [1] Making Large Language Models A Better Foundation For Dense Retrieval. (Link: https://huggingface.co/BAAI/bge-reranker-large)

---

> ### Author Response · Authors · 2024-12-01
> **Kind Reminder from Authors**
>
> Thank you once again for your thoughtful feedback and active engagement during the discussion phase! We truly appreciate the time and effort you have dedicated to reviewing our work. As **the discussion phase concludes in two days**, we would like to highlight the updates and enhancements we have made to our submission based on your valuable feedback: (1) Deep Analysis of the Reverse U-Curve Phenomenon; (2) Comparisons with Reranking Methods; (3) Comprehensive Cost-Benefit Analysis; (4) Intermediate Reasoning Analysis; (5) Hard Negative Analysis; (6) Detailed Comparison with Existing Works; (7) Clarification of Reordering Strategies. We hope these updates address your questions and provide a clearer picture of the contributions and significance of our work. If there are any remaining concerns or additional points for discussion, we would be more than happy to engage further before the deadline. In light of these clarifications and improvements, we kindly request your consideration of a higher assessment for our submission. Your feedback has been instrumental in refining this work, and we are grateful for your thoughtful review.

---

### Official Review · Reviewer_RnDx · 2024-11-04

**Soundness:** 3
**Presentation:** 4
**Contribution:** 4
**Rating:** 6
**Confidence:** 4

**Summary:**

This paper studies how RAG systems are impacted by the advancement in long-context LLMs. The study empirically shows how the increase in the context-length (which allows more passages to be used) doesn't necessarily imply improved performance of the system. They show that the performance first improves before declining with increases number of retrieved passages. This is attributed to the presence of "hard negatives" which confuses the LLMs. The authors next propose 3 ways to counter the "hard-negatives" issue. The first is the reordering of the retrieved passages and it is shown that it can be an effective non-training based method. The other two approaches are training-based: finetuning with explicit hard-negatives, and finetuning with an additional reasoning step to train the LLM to learn to reject hard negatives. They show how these approaches help improve performance. A well-performed set of ablations help see the impact of data distribution match, retriever choice, and context length of LLMs during training.

**Strengths:**

The paper focuses on answering a question that hasn't received rigorous attention yet: understanding if retrieval optimizations are not needed anymore in RAGs system if we have long-context LLMs. They approached it quite methodically by first trying to answer the question by looking at multiple LLMs (each with its own context length limit) and multiple retrievers and seeing the impact of the RAG system with increasing retrieved passages. The problem has lot of moving parts: LLM, context length limit, # of retrieved passages, choice retriever, evaluation and training datasets, to name a few. The authors have done a reasonably well job to balance these different parts to answer the question.

**Weaknesses:**

There are a few places where the observations can be better represented and new hypotheses can be developed and evaluated. Some examples are below (and in the "questions" section below):
- In Fig 1, the authors claim that for strong retriever, the performance either reduces or reaches a plateau with increasing number of retrieved passages whereas it is different for weak retriever. However, I don't see any difference from the figure. Even for BM25 (weak retriever) the performance has inverted U shape similar to e5 (strong retriever).
- The reordering algorithm assumes that hard negatives are typically lower in relevance and hence reordering allows them to be in the middle, therefore allowing the LLM to not get distracted by them. However, that's not necessarily true. They most likely would be at the top and hence are called "hard" negatives.
- It was unclear until we look at the appendix on what was the knowledge base used for the experiments. It is also not clear (maybe my mistake) on what was the retriever used for experiments in Sec. 5.
- A key issue in the experiments is that for all the datasets used, the knowledge base is wiki which is already a part of the LLMs pretraining data. So some of the observations made could be impacted by how much of the knowledge for answering the query is already in the LLM's weights and how much is from the retrieved passages. Since the answers to the query are most probably in the model weights due to pretraining data, the observations from the experiments could be flawed and we might not be able to make fair conclusions. It would be good if we can run these experiments with a knowledge base use for retrieval be something that is NOT a part of the LLM's pretraining data.

**Questions:**

- Fig 1 also has an interesting observation where the Gemini-1.5-pro is quite robust to increasing number of retrieved passages and doesn't decline in performance. Is there a correlation between the context windows of these long-context LLMs and their ability to be robust to increasing number of retrieved passages? And if so, would "infinite"-context window LLMs have most robustness? It would be interesting to perform additional study around this.
- The above is especially the case when answering the subsequent questions in the paper regarding the hard negatives. Since LLMs are not perfect processing blocks. Just because the correct one is present doesn't mean the LLM will get it right, and it depends on how good the LLM is at processing the given information.
- Fig. 7b indicates that there might not be much impact on the choice of the retriever. More information on that would be helpful.

---

> ### Author Response · Authors · 2024-11-20
> **Author Response (1/2)**
>
> We appreciate your insightful feedback and believe it has significantly strengthened our manuscript. We have carefully addressed each of your comments as detailed below:
>
> - **Difference between e5 curve and bm25 curve in Figure 1.**
>
> You correctly observe that the performance curves exhibit an inverted U shape with both strong (e5) and weak (BM25) retrievers. However, we emphasize that there are key distinctions between the two. As shown in Figure 1, the strong retriever (e5) generally achieves peak performance earlier (topk=10 for both Gemma-7B-chat and Gemma-2-9B-Chat) and then declines more sharply compared to the weak retriever (BM25, which peaks at topk=30).
>
> This behavior, as explained in Sections 3.2 and 3.3, stems from the fact that irrelevant passages retrieved by the stronger e5 retriever have a more detrimental impact on performance. We have further clarified this observation in the "Insights" paragraph of Section 3.1 to highlight this important nuance.
>
> - **It is possible that hard negatives are ranked high.**
>
> You raise a valid point about the potential for certain hard negatives to be ranked highly by the retriever. While this can occur in individual cases, our analysis demonstrates that, on average, hard negatives are ranked lower than relevant passages.
>
> To illustrate this, we conducted experiments on the NQ dataset with both e5 and BM25 retrievers, analyzing the top-k retrieved passages (k = 5/10/20/50/100). We calculated the average ranking of both relevant passages and hard negatives, first averaging at the query level and then across all queries.
>
> e5
> | # retrieved psg (top-k)          | 5     | 10    | 20     | 50     | 100    |
> |----------------------------|-------|-------|--------|--------|--------|
> | Avg. relevant psg ranking  | 2.497 | 4.171 |  7.406 | 16.638 | 30.826 |
> | Avg. hard negative ranking | 3.426 | 6.157 | 11.429 | 27.044 | 52.801 |
>
> BM25
>
> | # retrieved psg (top-k)          | 5     | 10    | 20     | 50     | 100    |
> |----------------------------|-------|-------|--------|--------|--------|
> | Avg. relevant psg ranking  | 2.696 | 4.808 |  8.609 | 19.615 | 36.252 |
> | Avg. hard negative ranking | 3.189 | 5.784 | 10.931 | 26.262 | 51.719 |
>
> Our findings consistently show that the average ranking of relevant passages is higher than that of hard negatives for both retrievers and across all top-k values. This observation supports the rationale behind our retrieval reordering strategy: by prioritizing higher-ranked passages, we increase the likelihood that the LLM focuses on relevant information.
>
> Furthermore, it's important to emphasize that the benefits of reordering extend beyond mitigating the impact of hard negatives. By placing higher-relevance passages in more prominent positions, we simultaneously reduce the influence of lower-relevance passages, leading to a more effective utilization of the retrieved information by the LLM.
>
> - **Knowledge base and the retriever for Sec. 5.**
>
> To enhance clarity and reproducibility, we have updated Section 5 to explicitly state that we use Wikipedia as the knowledge base and e5 as the retriever.
>
> - **Experiment on a corpus that is not in the pretraining data.**
>
> You raise a valid point regarding the potential overlap between LLM training data and Wikipedia. While the exact composition of these pretraining datasets remains undisclosed, it is important to note that even with potential overlap, LLMs often struggle to answer questions accurately without external knowledge (as demonstrated in Figure 5 when no passages are retrieved). Therefore, following established practices in RAG evaluation [1][2][3], we believe using Wikipedia as an external knowledge source remains a reasonable choice.
>
> However, we acknowledge the value of exploring diverse knowledge sources to strengthen our findings. To address your suggestion, we have expanded our analysis to include RAG experiments on PubMedQA [4] and BioASQ [4], utilizing PubMed [5] as the retrieval corpus.
>
> PubMedQA
> | # retrieved psg  | 0 | 1 | 5  | 10  | 20  | 30 | 40  | 50 | 100 | 150 |
> |---|---|---|---|--|---|---|---|---|---|---|
> | llama3.1-8B-chat  | 0.522 | 0.788 | 0.746 | 0.764 | **0.788** | **0.786** | 0.764 | 0.766 | 0.754 |  0.73 |
> | mistral-Nemo-Instruct | 0.314 | **0.592** |  0.41 |  0.43 |  0.46 | 0.416 |  0.38 | 0.394 |  0.32 | 0.354 |
>
> BioASQ
> | # retrieved psg  | 0  | 1  | 5  | 10  | 20  | 30  | 40  | 50   | 100  | 150  |
> |----|---|---|---|--|---|--|--|---|---|---|
> | llama3.1-8B-chat  | 0.7443 | 0.8414 | 0.8867 | **0.8997** | **0.9061** | 0.8819 | 0.8851 | 0.8819 | 0.8657 | 0.8511 |
> | mistral-Nemo-Instruct | 0.7524 | 0.7638 | **0.7767** | **0.7735** | 0.7330 | 0.7120 | 0.6974 | 0.6505 | 0.6845 | 0.6942 |
>
> These new experiments confirm the presence of the "inverted U-curve phenomenon" with both llama3.1-8B-chat and mistral-Nemo-Instruct as the generator across both datasets (with PubMed as the knowledge base and e5 as the retriever), further reinforcing our observations.

---

> ### Author Response · Authors · 2024-11-20
> **Author Response (2/2)**
>
> - **Is there a correlation between context windows of LLMs and their robustness in increasing the number of passages? Would "infinite"-context window LLMs have the most robustness?**
>
> This is an insightful observation, and we appreciate you bringing it to our attention. To investigate the potential correlation between LLM context windows and their robustness to increasing numbers of retrieved passages, we conducted experiments on the NQ dataset using two groups of LLMs with varying context window sizes: (1) 8K window LLMs: llama3-8B-Chat, gemma-2-9B-Chat, and llama3-70B-Chat; (2) 128K window LLMs: llama3.1-8B-Chat, mistral-nemo-12B-Instruct and llama3.1-70B-Chat.
>
>
> | # retrieved psg | 1  | 5 | 10 | 20 | 30  | 40  | 50  | 100 | 150 | 500 |
> |----|--|--|---|--|---|---|---|---|---|--|
> | llama3-8B-Chat (8k)  | 0.4432 | 0.5166 | 0.5161 | **0.5199** | 0.5025 | 0.4681 | - | - | - | - |
> | gemma-2-9B-Chat (8k)| 0.4687 | 0.5296 | **0.5435** | 0.5421 | 0.5271 | 0.5166 | - | - | - | - |
> | llama3-70B-Chat (8k) | 0.4454 | 0.5360 |     0.5576 |     0.5742 | **0.5839** | 0.5706 | -  | -  | - | - |
> | llama3.1-8B-Chat (128k) | 0.4684 | 0.5410 | **0.5474** | 0.5429 | 0.5355 | 0.5346 | 0.5343 | 0.5066 | 0.5025 | 0.4288 |
> | mistral-nemo-12B-Instruct (128k) | 0.4598 | 0.5355 | 0.5393 | **0.5593** | 0.5452 | 0.5474 | 0.5255 | 0.4825 |  0.4366 | 0.2856 |
> | llama3.1-70B-Chat (128k)  | 0.4789 | 0.5490 | **0.5607** | 0.5449 | 0.5429 | 0.5343 | 0.5427 | 0.5197 | 0.5186 | 0.4665  |
>
> Our findings indicate **no clear correlation between the context window size and robustness in increasing passage numbers.** Contrary to expectations, the smaller context LLM (llama3-70B-chat with 8k) exhibited greater robustness than the larger context LLM (llama3.1-70B-chat with 128k). Furthermore, models with the same maximum context window length (e.g., llama3.1-8B-chat and Mistral-Nemo-12B-Instruct) demonstrated significantly different robustness levels.
>
> These results suggest that factors beyond context window size, such as pretraining or post-training data, may play a crucial role in determining LLM robustness.  We hypothesize that despite supporting larger context windows, LLMs might exhibit a bias towards shorter contexts due to the prevalence of shorter training samples.  However, without access to the specific training data used for these LLMs, further investigation is needed to confirm this hypothesis. We acknowledge this limitation and recommend it as an area for future research.
>
>
> - **Fig. 7b indicates that there might not be much impact on the choice of the retriever.**
>
> We interpret these results from two key perspectives:
>
> **Retriever-Agnostic Improvement**: Our proposed RAG-specific tuning consistently enhances LLM performance across all retrievers, demonstrating the generalizability of our approach. This improvement is independent of the specific retriever used, highlighting the retriever-agnostic nature of our fine-tuning method.
>
> **Superiority of Mixed Hard Negatives**:  LLMs trained with RAG-specific data incorporating hard negatives from multiple retrievers outperform those trained with data from a single retriever. This observation underscores the importance of exposing the LLM to diverse hard negatives during training. It suggests that "hard negatives" exhibit retriever-specific characteristics, and training with a mix of retrievers equips the LLM to handle a wider range of challenging examples. This finding also hints that independent improvements in retrievers might have limited impact, emphasizing the importance of a joint end-to-end design of retrievers and LLMs in RAG systems.
>
> [1] RA-DIT: Retrieval-Augmented Dual Instruction Tuning. ICLR 2024.
>
> [2] Making Retrieval-Augmented Language Models Robust to Irrelevant Context. ICLR 2024.
>
> [3] RankRAG: Unifying Context Ranking with Retrieval-Augmented Generation in LLMs. NeurIPs 2024.
>
> [4] Benchmarking Retrieval-Augmented Generation for Medicine. ACL 2024.
>
> [5] PubMed: https://pubmed.ncbi.nlm.nih.gov/

---

> > ### Comment · Reviewer_RnDx · 2024-11-27
> >
> > Thank you for your response and the additional studies. I am not fully convinced that hard-negatives are "significantly" lower than the relevant passages that reordering impacts them differently. Even your experiments don't necessarily show that. I trust your empirical evidence that reordering helps but I don't think the reasoning is justified. If the average rank is so close between the relevant and hard negatives, reordering shouldn't impact them differently.
> >
> > I appreciate your work on showing results for alternate retrieval corpus (besides wiki), and suggest you put that in the paper atleast in the appendix.
> >
> > Overall, I believe I have scored you accurately, and will hence keep my score as-is. All the best!

---

> > > ### Author Response · Authors · 2024-11-27
> > > **Round2 Author Response**
> > >
> > > Thank you for your thoughtful feedback and suggestions. We have incorporated the additional results into the Appendix R as recommended.
> > >
> > > Regarding your concern about the impact of retrieval reordering on relevant passages versus hard negatives, we respectfully clarify the following: (1) **Differences in average rank with larger top-k values**: While the average rank between relevant passages and hard negatives may appear close for smaller top-k values, our rebuttal experiments demonstrate that this difference becomes significant as top-k increases. This supports our hypothesis that prioritizing higher-ranked passages improves the likelihood of the LLM focusing on relevant information over hard negatives (especially for larger top-k as shown in Figure 4). (2) **Lost-in-the-middle phenomenon**: As shown in Figure 4, the reordering effect is particularly impactful for larger top-k values. Longer inputs exacerbate the lost-in-the-middle phenomenon, where the LLM struggles to attend to middle-ranked passages. Retrieval reordering mitigates this issue by ensuring relevant passages are prioritized at the input's beginning or end, where the LLM’s attention is typically stronger.
> > >
> > > We hope this additional clarification addresses your concern. Thank you for your constructive feedback and for recognizing the value of our work.

---

### Author Response · Authors · 2024-11-20
**General Author Response**

Dear Reviewers,

We greatly appreciate your valuable feedback and thoughtful suggestions. In response to your comments, we have made revisions to our paper, with all changes clearly highlighted in blue.

First, we thank the reviewers for noting the strengths of our paper, particularly:
- The problems tackled are important and well-motivated. (RnDx, k49Q)
- The paper is clearly written. (RnDx, vMLV, ua3n, k49Q)
- The analysis is comprehensive and insightful. (RnDx, vMLV)
- The proposed method is substantial and sound. (vMLV, k49Q)
- The empirical results are consistent, comprehensive, and convincing. (RnDx, vMLV, ua3n, k49Q)

We have carefully addressed each reviewer's comments and questions individually in our detailed responses. We encourage you to review these responses and welcome any further requests for clarification.  The revised manuscript incorporates all of the reviewers' suggestions, including enhanced explanations of our analysis, new experimental results, and further details on the experimental settings.

We have made the following key updates:
- [Abstract] We have clarified the motivation of this paper. (ua3n)
- [Section 1] We have clarified the motivation and the summarization of our contribution. (ua3n)
- [Section 2] We have included a more detailed analysis of related work, highlighting the distinctions and advancements of our approach. (ua3n)
- [Section 3.1] We have added a sentence explicitly highlighting the observed differences between strong and weak retriever models. (RnDx)
- [Section 5.1, 5.2] We have included a comprehensive description of the retrieval corpus and the specific retriever employed across our various experiments. (RnDx)
- [Section 6] We have added further illustrations of the takeaways from the “Influence of retrievers on generalization” study. (RnDx)
- [Appendix O] We have expanded the case study with a more detailed analysis of retrieval queries. (vMLV)
- [Appendix P] We have included a cost-benefit analysis of basic reranking and our proposed method. (vMLV)
- [Appendix Q] We have added an analysis of the intuition behind retrieval reordering. (RnDx)
- [Appendix R] We have expanded our analysis to include RAG experiments on medicine domain problems. (RnDx)
- [Appendix S] We have added a section to discuss the potential future works. (vMLV)

In closing, we thank the reviewers again for your time and valuable feedback. If you have further concerns, please let us know, and we will be happy to address them.

---

### Meta-Review · Area_Chair_7RTC · 2024-12-20

**Metareview:**

This paper explores an important aspect of long-context large language models (LLMs), specifically how the quality of generated output initially improves but then declines as the number of retrieved passages increases. The study proposes a training-free method through retrieval reordering and two training-based methods: RAG-specific implicit LLM fine-tuning and RAG-oriented fine-tuning. Experiments conducted across multiple models and datasets provide insightful findings.

**Additional Comments On Reviewer Discussion:**

Discussion Summary During the Rebuttal Period:

1. Clarification: The motivation and the specific contributions of the study need further clarification to enhance the understanding of its significance.
2. Additional Experiments: The authors include experiments to validate the influence of the proposed methods in other domains, assess the effects of context window size, and perform comparative analysis using RetRobust.


This paper is well-motivated to me, and I recommend improving the statement of the goals and incorporating experiments related to this paper's focus in the final version.

---

### Decision · Program_Chairs · 2025-01-22

Accept (Poster)